# The Multifaceted Roles of Autophagy in Flavivirus-Host Interactions

**DOI:** 10.3390/ijms19123940

**Published:** 2018-12-07

**Authors:** Po-Yuan Ke

**Affiliations:** 1Department of Biochemistry & Molecular Biology and Graduate Institute of Biomedical Sciences, College of Medicine, Chang Gung University, Taoyuan 33302, Taiwan; pyke0324@mail.cgu.edu.tw; Tel.: +886-3-211-8800 (ext. 5115); Fax: +886-3-211-8700; 2Liver Research Center, Chang Gung Memorial Hospital, Taoyuan 33305, Taiwan; 3Division of Allergy, Immunology and Rheumatology, Chang Gung Memorial Hospital, Taoyuan 33305, Taiwan

**Keywords:** *Flaviviridae*, Autophagy, selective autophagy, Hepatitis C virus, Dengue virus, Japanese encephalitis virus, West Nile virus, Zika virus

## Abstract

Autophagy is an evolutionarily conserved cellular process in which intracellular components are eliminated via lysosomal degradation to supply nutrients for organelle biogenesis and metabolic homeostasis. Flavivirus infections underlie multiple human diseases and thus exert an immense burden on public health worldwide. Mounting evidence indicates that host autophagy is subverted to modulate the life cycles of flaviviruses, such as hepatitis C virus, dengue virus, Japanese encephalitis virus, West Nile virus and Zika virus. The diverse interplay between autophagy and flavivirus infection not only regulates viral growth in host cells but also counteracts host stress responses induced by viral infection. In this review, we summarize the current knowledge on the role of autophagy in the flavivirus life cycle. We also discuss the impacts of virus-induced autophagy on the pathogeneses of flavivirus-associated diseases and the potential use of autophagy as a therapeutic target for curing flavivirus infections and related human diseases.

## 1. Introduction

Autophagy is a lysosome-mediated catabolic process in which unwanted intracellular components are degraded to recycle nutrients for the regeneration of organelles and energy [1,2]. A variety of stresses, including nutrient starvation, organelle damage, accumulation of unfolded proteins and pathogen infection, have been demonstrated to activate autophagy [3,4]. Dysregulation of autophagy has been shown to play a role in the pathogeneses of numerous human diseases, such as cancer, neurodegenerative diseases and pathogen infection [5,6]. An increasing number of studies indicate that microbial infection activates autophagy, which degrades the invading microorganism and induces the innate immune response, thus restricting pathogen infection [7,8,9,10]. On the other hand, autophagy has emerged as a pro-viral pathway by which viruses activate to benefit viral growth in infected cells [11,12,13]. Approximately seventy enveloped, positive-strand RNA viruses belonging to the *Flaviviridae* family, including hepatitis C virus (HCV), dengue virus (DENV), Japanese encephalitis virus (JEV), West Nile virus (WNV) and Zika virus (ZIKV), that contain important human pathogens and thus exert a global burden on public health [14,15]. Flavivirus infections often induce rearrangement of the host cellular membrane to establish a membranous structure for viral growth, thus triggering a variety of stress responses in infected cells [16,17]. In recent years, emerging lines of evidence have shown that autophagy is activated by flavivirus infections to promote viral replication and counteract virus-induced stresses, such as protein unfolding and organelle damage [16,17]. Therefore, comprehensively understanding the interplay between host autophagy and flavivirus infection will provide key information for curing viral pathogenesis. In this review, we summarize the current knowledge on how *Flaviviridae* viruses manipulate and subvert host autophagy in infected cells and address the functional role of autophagy in flavivirus-host interactions as well as in the pathogeneses of virus-associated human diseases. Finally, we discuss the therapeutic potential of autophagy modulation in the intervention of flavivirus infection.

## 2. Autophagy

Autophagy is referred to as the “self-digestion” process in which eukaryotic cells sequester intracellular materials within double-membraned vesicle-like structures and then deliver these materials to lysosomes for degradation. At least three types of autophagy have been identified, including macroautophagy, microautophagy and chaperone-mediated autophagy (CMA) (Figure 1). Macroautophagy (hereafter referred to as autophagy) involves the delivery of cytoplasmic components to lysosomes for degradation via autophagic vacuoles [2,18]. Microautophagy involves the direct transport of cytosolic content to lysosomes via the invagination and scission of lysosomal membranes into the lumen [19,20]. CMA is a selective elimination process in which substrates containing the “Lys/Phe/Glu/Arg/Gln” (KFERQ) motif are recognized by a chaperone protein, heat shock cognate protein of 70 kDa (Hsc70) and delivered to the lysosomal lumen via the lysosomal membrane protein 2A (LAMP2A) [21,22]. Autophagy was discovered in the mid-1950s by Christine de Duve, the 1974 Nobel Laureate in Physiology or Medicine and coined in the early 1960s at the Ciba Symposium on Lysosome [23,24]. In addition, ultrastructural transmission electron microscopy (TEM) analyses demonstrated the morphogenesis of dense bodies with sizes similar to that of mitochondria in various types of tissues of animals [25,26,27,28]. These dense bodies morphologically represent double-membranous vesicle structures in which mitochondria and the endoplasmic reticulum (ER) are engulfed [26,28,29]. Shortly thereafter in the late 1970s, several groups showed that deprivation of amino acids and growth factors can induce autophagy [30,31]. Biochemical studies performed in the 1970s–1990s demonstrated that autophagy promotes the degradation of long-lived proteins, which correlates with a decreased supply of amino acids [32,33]. Meanwhile, several groups characterized the intracellular signaling processes and molecules associated with autophagy and identified 3-methyladenine (3-MA) as an autophagy inhibitor [34,35,36,37,38,39,40,41,42]. Moreover, the concept of a phagophore developed for double-membranous autophagic vacuoles was first described [43,44,45]. Yoshinori Ohsumi initiated the comprehensive identification of genes functioning in autophagy in his work to characterize autophagic vacuoles in *Saccharomyces cerevisiae* and genetically screen temperature-sensitive mutants that are defective in the formation of autophagic vacuoles in yeast cells [46,47,48]; Yoshinori Ohsumi was awarded the 2016 Nobel Laureate in Physiology or Medicine for his work. Approximately fifteen autophagy-related genes were identified as being required for the completion of autophagy in yeasts [47]. At the same time, several groups also uncovered autophagy-related genes (ATGs) functioning in humans and other eukaryotes and the nomenclature for ATGs among different eukaryotic species was unified [49,50,51,52,53]. Approximately forty ATGs have currently been identified, most of which are highly conserved in nearly all eukaryotes [53,54,55].

### 2.1. The Biogenesis of Autophagic Vacuoles

The stepwise process of vacuole biogenesis in autophagy initiates with an intracellular membrane rearrangement promoting the emergence of an isolation membrane (IM)/phagophore (Figure 1) [56,57,58]. The membranous structure of the IM/phagophore originates from many types of organelles, including the ER [59,60], mitochondria [61], Golgi apparatus [62], plasma membrane [63], recycling endosome [64,65] and mitochondria-associated ER membrane [66]. The newly formed phagophore subsequently elongates and encloses into a double-membranous vacuole, termed autophagosome (Figure 1) [67,68,69,70]. Then, the autophagosome fuses with lysosomes (Figure 1), generating mature autolysosomes that degrade the sequestrated cargo by acidic proteases [69,71,72,73]. The entire autophagic process relies on the coordinated actions of ATGs to rearrange membranes for vacuole biogenesis as well as on multiple cell signaling pathways [18,74,75]. When cells are starved of nutrients, autophagy is activated by the repression of mammalian target of rapamycin (mTOR), a serine/threonine protein kinase required for metabolic regulation. The suppression of mTOR induces translocation of the unc-51 like-kinase (ULK) complex, which is organized by ULK1/2, ATG13, RB1-inducible coiled-coil 1 (RB1CC1, also known as FIP200) and ATG101, from the cytosol to ER membrane-reconstituted compartments (Figure 1) [76,77]. Subsequently, the class III phosphatidylinositol-3-OH kinase (PI3K) complex (class III-PI3K, including Vps34/PI3KC3, Vps15, Beclin 1 and ATG14) is recruited to the ER-derived nucleation site, triggering the generation of phosphatidylinositol-3-phosphate (PtdIns(3)P) (Figure 1) [75,78,79]. The newly formed PtdIns(3)P, in turn, recruits the double-FYVE-containing protein 1 (DFCP1) and WD-repeat domain PtdIns(3)P-interacting (WIPI, the mammalian orthologue of ATG18) family proteins to generate an ER-associated omegasome structure, also known as the IM/phagophore (Figure 1) [80,81]. Elongation and enclosure of the IM/phagophore into a mature autophagosome relies on two ubiquitin-like (UBL) conjugation systems (Figure 1) [82,83,84,85]. ATG7 (E1) and ATG10 (E2) confer the conjugation of ATG12-ATG5, which interacts with ATG16L to form the ATG12-ATG5-ATG16L trimeric complex (Figure 1) [82,86]. At the same time, the phosphatidylethanolamine (PE)-conjugation of ATG8 family proteins (including the microtubule-associated protein 1 light chain 3 (LC3) and gamma-aminobutyric acid receptor-associated protein (GABARAP) subfamilies) initiates with the proteolytic cleavage of their C-termini by ATG4 family proteins [87,88]. Then, the cleaved ATG8 family proteins are conjugated to PE via the catalytic cascade of ATG7 (E1) and ATG3 (E2), producing the lipidated forms of ATG8 family proteins [87,88]. Finally, autophagosomes fuse with lysosomes to form mature autolysosomes (Figure 1), in which the sequestrated materials are degraded and recycled for nutrients.

The fusion of autophagosomes with lysosomes is also coordinated by multiple protein-protein interactions, cytoskeleton-mediated transport and membrane rearrangement events [70,71,73] (Figure 1). This fusion step initiates with the timely transport of autophagosomes and lysosomes via microtubules [89,90]. The small GTPase Ras-related protein 7 (Rab7) located on the surface of the autophagosome bridges the movements of the autophagosome on microtubules by binding to FYVE and coiled-coil domain-containing 1 (FYCO1) and Rab-interacting lysosomal protein (RILP) [91], which are respectively linked to kinesin and dynein on the microtubules [92,93,94,95]. Moreover, Rab7 located on late endosomes and lysosomes could be recruited to mature autophagosomes, thus promoting autophagosome-lysosome fusion [92,93]. The PI3K protein complex associated with UV radiation resistance-associated (UVRAG) also participates in the fusion of autophagosomes and lysosomes by interacting with Vps16, a subunit of the homotypic fusion and protein sorting (HOPS) complex [79,96,97]. In contrast, the binding of Rubicon to the PI3K protein complex inhibits autophagosome-lysosome fusion [79]. A further study highlights that ATG14L binds to syntaxin 17 (STX17) and synaptosome-associated protein 29 (SNAP29) binary complexes on the surface of autophagosomes, promoting STX17/SNAP29/vesicle-associated membrane protein 8 (VAMP8)-mediated autophagosome-lysosome fusion [98]. Furthermore, the pleckstrin homology domain-containing protein family member 1 (PLEKHM1) protein that contains an LC3-interacting motif has been shown to concomitantly interact with Rab7/HOPS and LC3, facilitating the fusion of autophagosomes and lysosomes [99]. Although the functional ATGs and other cellular proteins involved in the autophagic process have been extensively studied and characterized, the detailed regulatory mechanism underlying the functional roles of ATGs at each step of autophagy requires further investigation.

### 2.2. Selective Autophagy

In addition to bulk and nonselective degradation, increasing evidence indicates that autophagy can selectively eliminate specific cargos, including organelles and proteins [100,101,102]. Selective autophagy initiates with the recognition of cargo that are highly polyubiquitinated via cargo receptors and delivers degraded cargos to autophagic machinery via the binding of cargo receptors to ATG8 family proteins located on the autophagosomal membrane of IM/phagophore (Figure 2) [91,103,104,105]. Several cargo receptors of selective autophagy, such as neighbor of BRCA1 (NBR1), calcium-binding and coiled-coil domain-containing protein 2 (Calcoco2, also known as NDP52), p62/sequestosome 1 (SQSTM1) and optineurin (OPTN), contain the LC3-interacting regions (LIR) required for interaction with ATG8 family proteins, thus promoting the engulfment of cargos into autophagosomes [91,105,106]. Despite this LIR-containing cargo receptor-mediated clearance of ubiquitinated substrates, several potential ATG8-interacting motifs (AIMs) and GABARAP-interacting motifs (GIMs) have been identified within ATGs and other cellular proteins [107,108,109,110]. Recently, the AIMs located within *Saccharomyces cerevisiae* ATG19 were shown to directly bind to ATG5, thus recruiting the ATG5-ATG12-ATG16L complex to cargo and stimulating the PE conjugation of ATG8 for closure of the autophagosome [111]. Whether LIRs, AIMs and GIMs participate in the completion of autophagy and recruitment of potential substrates to the selective autophagy process requires further study.

Selective autophagy plays a pivotal role in maintaining the integrity of intracellular organelles by degrading damaged organelles (Figure 2) [100,102,112]. The selective elimination of organelles, termed organellophagy, supplies recycled nutrients for the regeneration of mitochondria, peroxisomes, ER, lipid droplets (LDs), ribosomes, lysosomes and nuclei (Figure 2). The removal of mitochondria by selective autophagy, termed mitophagy, could be activated by hypoxia [113,114], accumulation of reactive oxygen species (ROS) [115,116,117] and mitochondrial depolarization [118,119,120]. The major route for clearing damaged mitochondria originates from the loss of PTEN-induced putative kinase 1 (PINK1) cleavage by presenilin-associated rhomboid-like protein (PARL) within the inner mitochondrial membrane and the inhibition of PTEN degradation via the ubiquitin-proteasome pathway [121,122]. The PINK1 accumulated on the outer mitochondrial membrane, in turn, phosphorylates ubiquitin and then recruits the ubiquitin E3 ligase Parkin [118,119,120,123,124,125]. Parkin, in turn, ubiquitinates the surface proteins on the outer mitochondrial membrane [118,119,120,123,126], triggering the recognition of cargo receptors for the removal of mitochondria by autophagy [127]. PINK1 specifically recruits Calcoco2/NDP52 and OPTN to mitochondria and subsequently induces the translocation of phagophore-generating factors, including DFCP1 and WIPI, for autophagosome maturation proximal to degradative mitochondria [127]. Notably, the TANK binding kinase 1 (TBK1)-mediated phosphorylation of p62/SQSTM1 at serine residue 403 and OPTN at serine residues 177, 473 and 513 are critical for promoting mitophagy [128,129,130]. In addition to the PINK1/Parkin-mediated mitophagy pathway, several outer mitochondrial membrane proteins, such as FUN14 domain-containing 1 (FUNDC1), BCL2/adenovirus E1B 19 kDa protein-interacting protein 3 (BNIP3), BCL2/adenovirus E1B 19 kDa protein-interacting protein 3-like (BNIP3L) and yeast ATG32, have been shown to mediate mitophagy in a ubiquitin-independent manner [131,132,133,134,135]. Not surprisingly, several new molecules were recently identified as new cargo receptors of mitophagy, including prohibitin 2 (PHB2) and Toll-interacting protein (Tollip) [136,137]. Reciprocally, the dequbiquitination enzymes USP30 and USP35 have been shown to antagonize the mitophagy process by deubiquitinating Parkin [138,139].

In addition to mitophagy, several types of organelles have been shown to be removed by selective autophagy (Figure 2). The cognate cargo receptors responsible for the clearance of these organelles have been recently identified. For pexophagy, known as the degradation of peroxisomes by selective autophagy, yeast ATG36 and mammalian NBR1 and p62/SQSTM1 function as adaptors for the recruitment of damaged peroxisomes to autophagic degradation machinery (Figure 2) [140,141,142,143]. Multiple types of kinases, such as yeast Hrr25 and mammalian Ataxia-telangiectasia-mutated (ATM), phosphorylate these cargo receptors, facilitating their interaction with the autophagosome membrane [144,145]. Moreover, polyubiquitination of several peroxisomal (PEX) membrane proteins, such as PEX5 and the 70-kDa PEX membrane protein (PMP70), serves as a signal for recognition by cargo receptors [145,146]. The degradation of stressed ER by ER-phagy relies on the functions of yeast ATG39, ATG11 and ATG40 [147] as well as on the family with sequence similarity 134, member B (FAM134B) and reticulon family proteins (Figure 2) [148,149]. Notably, ATG39 and ATG11 have also been shown to participate in the selective degradation of yeast nuclei (Figure 2) [147]. Selective autophagy also involves the clearance of protein aggregates through p62/SQSTM1 and histone deacetylase 6 (HDAC6)-mediated recognition of Lys63 (K63)-linked poly-ubiquitin chain on aggregated proteins [150,151,152,153] (Figure 2). In addition, NBR1 and the autophagy-linked FYVE (ALFY) are shown to cooperate with p62/SQSTM1 to eliminate protein aggregates [154,155,156,157]. Lysophagy was recently identified as a new pathway for removing injured lysosomes via the concerted recruitment of galectin-3 and LC3 onto lysosomal membranes, as these proteins are presumably recognized by p62/SQSTM1 and targeted for degradation via autophagy (Figure 2) [158,159]. Very recently, autophagy has emerged as a critical pathway for the degradation of ribosomes, termed ribophagy (Figure 2) [160,161]. In addition, autophagy has been shown to catabolize LDs to maintain metabolic homeostasis (Figure 2) [162,163]. CMA also regulates the degradation of LDs via the 5′-AMP-activated protein kinase (AMPK)-mediated phosphorylation of perilipin 2, promoting recognition of the CMA chaperone Hsc70 for degradation [164]. Moreover, the newly identified cargo receptor of autophagy, nuclear receptor coactivator 4 (NCOA4) interacts with ATG8 family proteins and promotes degradation of the ferritin heavy and light chains, thus controlling intracellular ion levels (Figure 2) [165,166]. This autophagy-mediated turnover of ferritin is termed ferritinophagy, which has been suggested to regulate erythropoiesis and DNA replication in blood cells [167,168]. Elimination of invading microbes acts as the first line of host defense to counteract pathogen infection [169,170,171]. To this end, in the xenophagy process, invading pathogens are directly engulfed and eliminated via the p62/SQSTM1-, Calcoco2/NDP52- and OPTN-mediated recognition of ubiquitinated microbial proteins (Figure 2) [172,173,174]. This process also requires the phosphorylation of p62/SQSTM1 (at serine residues 349 and 403) and OPTN (at serine 177) [174,175,176,177]. Furthermore, autophagy also participates in the activation of an innate immune defense response to inhibit microbial infection, as it induces Toll-like receptor (TLR)-mediated immune responses and facilitates the presentation of antigens derived from pathogens to major histocompatibility complex (MHC) class II molecules [178,179,180,181,182]. Collectively, autophagy not only regulates cellular homeostasis via selectively removing harmful organelles from cells but also serves as a host defensive mechanism to restrict pathogen infection.

### 2.3. Autophagy and Diseases

Autophagy is a catabolic pathway utilized by eukaryotic cells to counteract stress responses, promote organelle turnover and eliminate aggregated proteins and excess lipids [1,2]. Moreover, autophagy also degrades infectious microbes and participates in regulating the immune response [100,102,112,183]. Therefore, autophagy protects cells from damage and maintains cellular homeostasis [5,6,183]. Dysregulation of autophagy has been implicated in the development of various types of human diseases, including cancer [183,184], neurodegenerative diseases [185,186,187,188], infectious diseases [7,172], cardiovascular diseases [189,190,191], aging [184,185,189,190,191,192,193] and metabolic disorders [183,194,195,196,197]. Thus, the modulation of autophagy presents a valuable and potential therapeutic target for treating human diseases. However, multiple obstacles, including drug specificity, the limited in vivo models for testing drug potency and unexpected side effects, hamper the implementation of autophagy activators and inhibitors in the clinic. Comprehensively understanding the entire autophagic process will provide more information for the development of rationally designed autophagic modulators in the future.

## 3. Flaviviridae Viruses

The *Flaviviridae* family comprises enveloped single-stranded, positive-sense RNA viruses that are approximately 50~70 nm in diameter [14,15,198] and includes four major genera: *Flavivirus*, *Hepacivirus*, *Pestivirus* and *Pegivirus* [15,198,199]. Several members of the *Flavivirus* and *Hepacivirus* genera are human pathogens that cause millions of infections annually and thus exert a global burden on public health [14,15,198]. HCV, a member of the *Hepacivirus* genus, is a leading cause of chronic hepatitis and at least 3% of the human population is infected by HCV worldwide [200,201]. Long-term chronic HCV infection often leads to multiple liver diseases, including steatosis, cirrhosis and hepatocellular carcinoma (HCC) [202,203]. The four prevalent human pathogens DENV, WNV, JEV and ZIKV belong to the genus *Flavivirus* [199,204,205]. These mosquito-transmitted viruses account for most severe arbovirus infections in the human population and irregular outbreaks exert an enormous threat on global health [205,206,207,208,209,210,211,212]. Primary DENV infection often causes reactions ranging from asymptomatic illness to symptoms such as rush, headache and dengue fever [205,206,213]. Secondary, homo- or heterotypic infections may develop into severe dengue hemorrhagic fever (DHF) and dengue shock syndrome (DSS), most likely due to antibody-dependent enhancement (ADE), which increases viral infectivity and pathogenicity [205,206,213,214]. JEV is a major cause of encephalitis flavivirus in the Asia-Pacific region [207,208,215]. No significant symptoms are observed in the vast majority of JEV-infected individuals but severe clinical illness still occurs in rare cases of infection, resulting in an approximately 30% mortality rate, particularly in children [207,208,215]. Most remnant-infected patients who survive often suffer permanent neurological deficits, including seizures, paralysis and mental retardation [207,208,215]. WNV is an emerging and mosquito-transmitted flavivirus that causes fever and neuroinvasive diseases, including encephalitis and poliomyelitis [210,216] and serious illness in aged and immunocompromised patients often introduces a higher risk of mortality [217,218]. The recently re-emerged ZIKV led to large outbreaks in South and Central America from 2015–2016 [211,219]. The virulence of this re-emerged ZIKV strikingly increases the incidence rates of microcephaly in newborn infants and Guillain-Barré syndrome in adults and this virus thus serves as another threat to public health worldwide [220,221,222]. No approved vaccination strategies against these human pathogenic flaviviruses are available except for JEV [223,224]. Recently, oral direct-acting antiviral (DAA) drugs targeting HCV replication have been approved and shown to cure HCV infection in more than 90% of patients [225,226]. However, antiviral drugs for the clinical treatment of other flavivirus infections are still limited [227].

### 3.1. Hepatitis C Virus (HCV)

HCV was first discovered to be the infectious agent underlying non-A, non-B hepatitis in 1975 [228] and the HCV viral RNA sequence was cloned in 1989 [229]. Currently, seven isolates of HCV have been cloned and classified, including genotypes 1 through 7 and each genotype is further divided into an array of different subtypes [230]. The high genomic variability among the different HCV genotypes may lead to poor cross-genotype immunity and varying levels of disease progression [231,232]. In recent decades, the combined therapy of pegylated-interferon (IFN) and ribavirin has been standardized for treating HCV infection [233]. However, the genotype of the infecting virus, genetic polymorphisms, disease stage and severe side effects often hamper the successful rate of treatment [225,233,234]. IFN-free and DAA-based anti-HCV therapy has led to an HCV infection cure rate of more than 90% [225,226]. However, the emergence of resistance to DAA drugs [225,235] and the uncertain and controversial effects of DAA treatment on disease progression in chronically infected patients still impede the complete eradication of HCV infection [236,237]. For instance, data on the impact of DAA treatment on the risk for HCC occurrence in chronic HCV patients are still conflicting. Several studies have demonstrated that the rate of HCC development is significantly reduced in patients who have achieved a sustained virological response (SVR) after DAA therapy [238,239,240]. However, numerous reports have indicated that the rates of HCC recurrence and de novo development of HCC in patients after DAA-induced SVRR are unexpectedly increased [241,242], possibly due to uncontrolled liver immunity resulting from the DAA-mediated eradication of HCV-specific T cells [243].

The HCV viral genome comprises 9.6 Kb of positive-sense single-stranded RNA (ssRNA) that contains an open reading frame (ORF) and two untranslated regions (UTRs) located on the 5′- and 3′-termini [201,244,245,246]. The positive strand of the HCV RNA genome can be directly translated into a polypeptide of approximately 3300 amino acids and then processed into three structural proteins (core and envelope glycoproteins E1 and E2) and seven nonstructural (NS) proteins (p7, NS2, NS3, NS4A, NS4B, NS5A and NS5B) by a combination of cellular and viral proteases [201,244,246]. The core, E1 and E2 proteins are the major constituents responsible for assembly of the HCV virion [7,196,244,245], while the NS proteins are required for organization of the replication complex and reconstitution of membranous compartments, that is, a membranous web for HCV viral RNA replication [27,101,247,248]. Moreover, the concerted localization of NS proteins on the surface of LDs also participates in HCV viral particle assembly [249,250].

The entry of HCV into hepatocytes relies on several cell surface molecules, termed entry (co)receptors, including tetraspanin CD81 [251,252,253], scavenger receptor class B member 1 (SCRAB1) [254,255], tight junction proteins, claudin 1 (CLDN1) [256] and occludin (OCLN) [257]. Moreover, the lipoprotein-binding proteins, proteoglycan and leptin-binding protein located on the surface of hepatocytes, such as low-density lipoprotein receptor (LDLR) [258,259], heparan sulfate (HS) [260,261] and dendritic cell-specific intercellular adhesion molecule three grabbing nonantigen (DC-SIGN) [262], have been shown to facilitate the attachment of lipoprotein-associated HCV virions to the cell surface. Following the initial attachment to the cell surface, viral particles bind to SCARB1 and CD81 via interactions with the E1 and E2 structural proteins [263,264,265]. Then, the epidermal growth factor receptor (EGFR) and downstream effector GTPase HRas trigger the association of CD81 with CLDN1 to facilitate the binding of HCV to OCLN on the basolateral surfaces of hepatocytes [266,267,268,269,270]. The viral particle is subsequently internalized via a clathrin-mediated and pH-dependent endocytosis pathway [271,272]. After the viral envelope fuses with the endosomal membrane, HCV undergoes an uncoating process to release the viral genome into the cytoplasm, wherein viral RNA is translated and replicated [271,272]. Moreover, several additional molecules, such as ephrin receptor A2 (EphA2) [270], Niemann-Pick C1-like L1 (NPC1L1) cholesterol uptake receptor [273], transferrin receptor (TfR) [274], tetraspanin CD63 [275] and cell death-inducing DFFA-like effector B (CIDEB) [276], as well as two additional tight junction proteins, CLDN6 and CLDN9, could mediate the entry of HCV into peripheral blood mononuclear cells lacking CLDN1 expression [277,278]. Notably, the second extracellular loop of OCLN together with CD81 and CIDEB respectively contribute to the host and tissue tropism of HCV entry [257,276]. The detailed molecular mechanism by which these newly identified entry (co)receptors mediate HCV entry remains to be elucidated.

The development of HCV-related liver diseases is a complicated, long-term process that has not been completely elucidated. Primary HCV infection is often asymptomatic and self-limiting but severe acute hepatitis and fulminant hepatic failure do occur rarely [279,280,281,282]. Most HCV infections are persistent and 20~30% of chronically infected patients develop liver steatosis, fibrosis and cirrhosis [279,280,281,282]. Ultimately, 3~5% of infected individuals develop HCC [279,280,281,282,283,284,285,286]. In addition to liver-associated diseases, HCV chronic infection also leads to extrahepatic diseases, such as mixed cryoglobulinemia vasculitis [282,287,288] and is perhaps linked to the progression of metabolic disorders, that is, diabetes and insulin resistance [289,290].

### 3.2. Dengue Virus (DENV)

DENV, the major mosquito-borne human pathogen, infects approximately 3.6 billion people in more than 100 countries worldwide and is thus a public health problem [206,291,292,293]. Since DENV was first isolated by Ren Kimura and Susumu Hotta in 1943 [293,294], four major serotypes of DENV (DENV1-4) have been identified and their genomic sequences are at least 65% similar [295,296]. Transmission of DENV to vertebrate host cells relies on its natural hosts, the mosquito vectors *Aedes aegypti* and *Aedes albopictus* [275,276,296,297]. Most DENV primary infections are asymptomatic and self-limiting [187,188]. However, clinical symptoms ranging from subclinical infection to dengue fever (DF) to the most severe forms, severe DHF and DSS, threaten the lives of some DENV-infected patients [298,299,300]. DF is a predominant symptom after primary and secondary infections, accompanied by febrile seizures, headache, rash and retro-orbital pain lasting for 1~2 weeks [298,299,300]. DHF, a severe dengue symptom that often results from homologous reinfection or secondary infection by a different DENV serotype, is characterized by thrombocytopenia, liver injury and hemorrhagic manifestations that ultimately induce DSS, which may cause mortality [298,299,300]. To date, effective antiviral drugs that cure DENV infection and vaccinations against DENV infection are still unavailable and under developed [227,300,301,302].

The DENV virion, a spherical particle that is appropriately 50 nm in diameter, is enclosed by a lipoprotein envelope [295,297]. The DENV viral genome is approximately 11 kb in length and contains an ORF that encodes three structural proteins (capsid, pre-membrane (prM) and glycosylated envelope E) and seven NS proteins (NS1, NS2A, NS2B, NS3, NS4A, NS4B and NS5) [291,297]. Particularly, the E protein harbors the membrane fusion ability to help virions attach to the host cell surface, interact with entry factors and assemble viral particles [303,304]. DENV infection of the host cell initiates with E protein-mediated receptor binding followed by clathrin-mediated endocytosis [305,306]. The diverse molecules present on the cell surface, including HS [307], DC-SIGN [308,309], mannose receptor (MR) [310], the lipopolysaccharide (LPS) receptor CD14 [311], the heat shock proteins 70 (HSP70) and 90 (HSP90) [312], 78-kDa glucose-regulated protein (Grp78) [313], lectin [314,315,316], laminin receptor [317], T cell immunoglobulin domain and mucin domain (TIM) and Tyro3, Axl and Mertk (TAM) phosphatidylserine receptors [318,319], as well as the tight junction protein CLDN1 [320,321] mediate the attachment of DENV and facilitate the entry of the virus into cells. The broad expression panel of these entry-associated factors in different types of tissues implies the low specificity for DENV entry receptors and the wide range of DENV tissue tropism.

After entering host cells, the viral genome is then released from the late endosome into the cytosol [305,306] and used as a template for translation and viral replication. Notably, the nondegradative ubiquitination of the DENV capsid protein is required for viral genome uncoating [322]. The NS proteins involves in the replication of viral RNA and the structural proteins assemble with the nascent viral RNA to generate infectious particles [15,300,323,324]. In a similar fashion to HCV, the DENV capsid protein also destabilizes the ER-derived membranous structure and LDs for virion assembly [325,326,327]. Due to the lack of an antiviral drug with high potency and an available vaccine for effective intervention against DENV infection, more efforts focused on understanding the DENV-host interactions will promote the exploration of potential therapeutic targets and new drug development [227,300,301,302]. In addition, elucidation of DENV-induced host cellular responses is urgently needed to delineate the pathogenesis of DENV-associated diseases.

### 3.3. Japanese Encephalitis Virus (JEV)

JEV is an infectious agent that remarkably leads to severe neurological disorders, affecting individuals in approximately 25 Asian countries [212,215]. JEV infection is considered asymptomatic in most infected individuals but a spectrum of clinical symptoms ranging from acute febrile seizure onset to severe encephalitis can appear [207,208,215]. Acute encephalitis often causes persistent neurological damage and mental disorders, such as parkinsonian syndromes and flaccid paralysis [207,208,215]. Based on nucleotide sequence similarities in the viral genome, five genotypes of JEV have been identified and divided (I through V) and infections by each genotype are prevalent in different regions [328]. JEV, first discovered and isolated in 1935 [329], is enzootically transmitted among mosquitoes (the genus *Culex*) and vertebrate hosts, including wild wading birds, cattle and pig [212,215,328]. JEV is a small enveloped flavivirus that is 50 nm in diameter. The JEV genome comprises approximately 11 Kb of positive-sense ssRNA that contains an ORF flanked by 5′- and 3′-UTRs [212,215]. After virus entry into permissive cells, the viral RNA can be translated into an ~3400 amino acid polypeptide precursor, which is co- and post-translationally processed by host signal peptidases and viral proteases into three structural proteins (capsid, prM and envelope E) and seven NS proteins (NS1, NS2A, NS2B, NS3, NS4A, NS4B and NS5) [212,215,330,331]. JEV shares a viral entry feature common to most flaviviruses, as it relies on virion attachment and interactions of the E protein with entry factors on the host cell surface [332,333]. In a similar fashion to DENV infection, the entry of JEV into a wide spectrum of susceptible cells requires several host molecules, including glucoaminoglycans (GAGs) [334,335,336], C-Type lectins [337,338,339,340], integrins [341], HSP70 [342,343,344], Grp78 [345], CD14 [346] and the intermediate filament vimentin [347,348]. Several types of vaccination strategies have been used for the clinical intervention of JEV infection [215,349]. However, virus replication-targeted antiviral treatment other than the currently supportive care for JEV-induced encephalitis is still limited and urgently needs further development.

### 3.4. West Nile Virus (WNV)

WNV is a neurotropic, arthropod-borne flavivirus that is transmitted in an enzootic cycle between mosquitoes, birds and other vertebrates [210,350]. WNV was first discovered and isolated from a female patient in the West Nile district of Uganda in 1937 [351]. WNV is also a human pathogen that may lead to significant morbidity and mortality [210,350,352]. The WNV viral genome comprises positive-sense ssRNA that is ~11 Kb in length [210,350]. WNV genomic RNA is used as a template for the translation of a polypeptide, which is subsequently processed by a combination of cellular and viral proteases to generate three structural proteins (capsid, prM and envelope E) and seven NS proteins (NS1, NS2A, NS2B, NS3, NS4A, NS4B and NS5) [210,350]. Analogous to other members of the *Flavivirus* genus, the structural proteins confer the encapsidation of viral RNA to form a virion, whereas NS proteins participate in replication of the viral genome [210,350,352]. WNV infection of host cells depends on several entry factors, such as GAGs [353], DC-SIGN [354,355], C-type leptin [356] and integrin [357,358]. In humans, WNV infection is often asymptomatic but approximately 20% of infected individuals suffer symptoms that range from febrile seizures to severe neurological disorders [210,350,352,359]. In rare cases of WNV infection, severe neurological manifestations can lead to death [210,350,352,359]. Thus far, a vaccine for WNV intervention and clinically approved drugs for eradicating WNV infection remain in development.

### 3.5. Zika Virus (ZIKV)

ZIKV is a mosquito-borne human pathogen that was first isolated from humans in 1954 [360]. ZIKV is an enveloped virus that harbors a positive-sense ssRNA that is ~10 Kb in length [211,361,362]. The viral RNA genome contains UTRs at the 5′- and 3′-termini and an ORF [14,15,198,209,221]. The viral RNA can be used as a template for translation of a polypeptide that is further cleaved by host and viral proteases into three structural proteins (capsid, prM and envelope E) and seven NS proteins (NS1, NS2A, NS2B, NS3, NS4A, NS4B and NS5) [162,167,195,220,261]. The structural proteins participate in assembly of the virion, while the NS proteins are responsible for viral RNA replication, viral-encoded nascent polypeptide processing and modulating the host cellular response [123,187,304,317,354]. Because the amino acid sequences of the DENV E and ZIKV E proteins are more than 50% similar, ZIKV infection may use a similar route and entry factors, such as TAM phosphatidylserine receptor [363,364] and GAGs [365], function in the entry of DENV into host cells [332,366]. ZIKV infection in humans is considered an asymptomatic illness but 10~20% of infected patients develop flu-like symptoms 7~14 days after infection [14,15,198,209,221]. However, this rate was enhanced by nearly 50% among individuals infected by the ZIKV that re-emerged in America from 2013~2015, characterized by febrile seizures, organ damage, encephalitis and thrombocytopenia [14,15,198,209,221,367]. Most importantly, ZIKV infection in pregnant patients leads to microcephaly and congenital ZIKV syndrome in newborn infants [14,15,198,209,221,367]. Notably, ZIKV infection in adults results in Guillain-Barré syndrome [368,369,370,371,372,373,374,375]. Antiviral drugs for curing ZIKV are still unavailable but multiple kinds of vaccines have been initially developed and are anticipated to combat ZIKV infection [376,377,378,379,380].

## 4. Flavivirus-Autophagy Interactions

The virus-induced modulation of autophagy was first discovered in the mid-1960s [381]. Ultrastructural electron microscopy analysis revealed immense membraned-enclosed bodies in poliovirus-infected HeLa cells [381]. The authors also showed that these virus-induced double-membraned bodies associate with virions [381]. Soon after this study, these vesicle-like structures associated with lysosomal enzymes were extensively observed in picornavirus- and herpesvirus-infected tissues and cells [382,383,384,385,386]. During the 1990s–2000s, several studies demonstrated that poliovirus infection induces autophagic process to support viral-induced membranous structure formation [387,388]. Later, Levine and colleagues first identified that the herpes simplex virus-1 (HSV-1) neurovirulence protein ICP34.5 can antagonize elongation initiation factor alpha (eIF2 alpha) kinase (also known as PKR)-mediated autophagy activation [389]. Prentice et al. showed that coronavirus mouse hepatitis virus (MHV) induces autophagy to promote the formation of a replication complex for MHV growth [390]. Christian Münz’s group demonstrated that autophagy participates in the delivery of Epstein-Barr virus nuclear antigen 1 (EBNA1) to lysosomes, contributing to the processing of MHC class II molecules [179]. The suppressive effect of autophagy inhibition via 3-MA on viral replication was also demonstrated in poliovirus- and rhinovirus-infected cells [391]. Human parvovirus B19 was also shown to activate autophagy to protect infected cells from viral-induced cell death [392]. In contrast to its pro-viral function, autophagy was also shown to play an antiviral role by inhibiting virus infection in the mid-2000s. Talloczy et al. provided the first line of evidence that xenophagy can eliminate HSV-1 via the eIF2alpha-PKR signaling pathway [393]. In addition to supporting virus replication and degrading the infecting virus, autophagy also contributes to the apoptosis of CD4+ T lymphocytes after human immunodeficiency virus (HIV) envelope proteins bind to C-X-C chemokine receptor type 4 (CXCR-4) in infected cells [394], suggesting that virus infection can trigger the HIV-induced destruction of CD4+ T cells. On the other hand, autophagy is also thought to mediate the recognition of ssRNA viruses and the production of IFN by plasmacytoid dendritic cells (pDCs) via TLR7 [178]. These results collectively suggest that autophagy plays diverse roles in virus-host interactions by promoting viral replication via supporting membranous compartments, degrading infectious viruses via xenophagy, enhancing antiviral immunity by enhancing MHC-mediated antigen presentation, protecting infected cells from death and triggering viral infection induced by the elimination of T lymphocytes [9,11,13,395,396,397]. In the late 2000s, numerous studies demonstrated that autophagy is induced by *Flaviviridae*, among which HCV and DENV were the first shown to activate autophagy and thus benefit the viral life cycle [398,399,400]. To date, nearly one hundred studies have investigated the role of flavivirus-infected autophagy in the viral life cycle and host cell responses. These studies indicate that flavivirus-activated autophagy may participate in the replication of viral RNA, translation of viral RNA, the entry of virions and assembly of infectious particles. Moreover, autophagy also participates in modulation of the virus-induced antiviral immune response and elimination of organelles and proteins in flavivirus-infected cells. On the other hand, flavivirus infection also induces autophagy to trigger the xenophagy pathway and thus degrade infecting viruses. Although autophagy plays a key role in flavivirus-host interactions, controversy and discrepancies exist among studies. Thus, we comprehensively summarize the current knowledge of the interplay between flaviviruses and host autophagy and the functional impacts on virus replication and host response. We also discuss the molecular mechanism by which flaviviruses drive autophagy to promote their life cycle and the pathogeneses of flavivirus-related diseases.

### 4.1. HCV

In the past decade, numerous studies have extensively shown that HCV modulates autophagy to promote the viral life cycle and alter cellular signaling. Ait-Goughoulte and colleagues first showed that serially passaging HCV genotype 1a (clone H77) in immortalized human hepatocytes (IHH) induces autophagic vacuole formation, as shown by TEM analysis and enhanced the generation of GFP-LC3-labeled autophagosomes, as determined by immunofluorescence (IF) assays (Table 1) [399]. Not only was the number of autophagic cells increased but the levels of Beclin 1 and ATG12-ATG5 conjugate were also elevated in HCV/H77-replicated IHH cells [399]. Soon afterward, transfection of a full-length HCV genotype 2a JFH1 viral RNA into human hepatocellular carcinoma Huh7 cells was reported to trigger incomplete autophagy, as demonstrated by the increased formation of autophagosomes rather than autolysosomes, inhibition of autophagic flux and degradation of long-lived proteins (Table 1) [401]. In addition, the authors also demonstrated that the unfolded protein response (UPR) is required for HCV replication-induced incomplete autophagy [401]. Moreover, their study also indicated that individually knocking down UPR- and autophagy-related genes significantly repressed the replication of HCV viral RNA [401]. To understand whether HCV JFH1 infection indeed induces autophagy in Huh7 cells, Chisari’s groups first analyzed the impact of cell culture-derived infectious HCV (HCVcc) on host autophagy in Huh7.5.1 cells, a clone derived from Huh7 cells that is highly permissive to HCV replication [402,403,404]. They first demonstrated that HCVcc JFH1 infection can induce the autophagy required for translation of the incoming viral RNA to establish virus infection (Table 1) [405]. Moreover, no colocalization of HCV NS proteins with viral-induced GFP-LC3-labeled autophagic vacuoles was observed, suggesting that the HCV-induced autophagic membrane does not primarily provide the replication compartment for HCV replication [405]. Ke and Chen demonstrated that HCVcc JFH1 infection induces the entire autophagic process throughout the formation of mature autolysosomes [405,406]. This study showed that HCV infection enhances autophagosome and autolysosome formation and increases autophagic flux [405,406]. Moreover, interference with autophagy using gene silencing and pharmacological inhibitors strikingly inhibits HCV viral RNA replication rather than the translation of viral RNA [405,406]. Most importantly, the HCV pathogen-associated molecular pattern-mediated IFN response was elevated by repressing HCV-induced autophagy in infected cells, suggesting that HCV may induce complete autophagy to suppress innate antiviral immunity [405,406]. Consistent with this study, Shrivastava et al. reported that genetically silencing ATGs, such as Beclin and ATG7, reduces HCV infectivity and activates IFN-stimulated gene expression in HCV H77-infected IHH cells (Table 1) [407]. This conclusion was further strengthened by a recent study showing that HCV-induced autophagy degrades tumor necrosis factor receptor (TNFR)-associated factor 6 (TRAF6) via p62/SQSTM1, suppressing host innate immunity (Table 1) [408]. Collectively, these studies indicate that HCV-activated autophagy may suppress antiviral innate immunity to promote HCV replication [405,406,407,409].

Apart from repressing the antiviral response to support viral replication, several studies have suggested that HCV-induced autophagy may enhance viral RNA replication via interactions between autophagic machinery and viral proteins (Table 1) [411,412]. Guevin and colleagues found that ATG5 transiently interacts with NS5B and NS4B during the early stage of HCV infection and that silencing ATG5 downregulates the intracellular amount of viral RNA in infected cells, implying that autophagy promotes the initial replication of nascent viral RNA by affecting NS5B polymerase activity and the NS4B-altered membranous web, which are necessary for HCV replication [412]. Another study further showed that the HCV NS proteins-organized replication complex along with viral RNA localize within the HCV-induced autophagosome [411], suggesting that HCV-activated autophagy provides a resource for reconstituting the membrane compartment to replicate HCV viral RNA. In agreement with this study, Sir et al. showed that HCV NS5A, NS5B and HCV viral RNA colocalized with GFP-LC3-labeled autophagosomes by IF analysis [411]. These authors further demonstrated that HCV viral RNA localizes to the autophagosome membrane using coimmunoprecipitation and immunogold-TEM assays [411], in agreement with another study showing that HCV may activate autophagy to promote the biogenesis of double-membraned vesicles (DMVs) that support the replication of HCV viral RNA (Table 1) [413]. Very recently, a biochemical fraction study demonstrated that HCV infection triggers the translocation of lipid rafts to autophagosomes to promote viral replication [414]. In contrast, Bartenschlager′s research group demonstrated no significant colocalization between autophagic vacuoles and HCV NS proteins or the dsRNA replicative intermediate [435] and that the generation of HCV-induced DMVs are associated with ER and LDs rather than with autophagic vacuoles in HCVcc-infected and replicon cells [436]. These results coincide with those of other studies showing that HCV-triggered autophagic vacuoles do not colocalize with HCV replication complex components [405,437]. These studies imply that the viral-induced autophagic process is not required for organization of the HCV viral RNA replication platform.

Autophagy was also shown to promote HCV virion assembly. Tanida et al. showed that knockdown of ATG7 and Beclin 1 moderately reduces the extracellular infectivity of infected cells without affecting the intracellular expression of viral proteins and RNAs (Table 1) [415], implying that HCV-induced autophagy may help the egress and release of mature virions. This observation was further supported by Shrivastava’s study, which showed that HCV-activated autophagy participates in the budding of infectious viral particles via the CD63-associated exosome pathway (Table 1) [416]. Moreover, Kim and Ou showed that HCV induces autophagy to regulate apolipoprotein E (ApoE) transport and thus promote HCV virion assembly (Table 1) [417].

Furthermore, autophagy was shown to be differentially activated in a genotype-dependent manner in HCV replicon cells, supporting viral replication rather than the complete viral life cycle (Table 1) [418]. Additionally, the replication of HCV Con1 (genotype 1b) replicon RNA interferes with autophagy maturation and impedes the secretion of cathepsin upon autolysosome maturation [418]. Moreover, inhibiting the HCV Con1 replicon-induced autophagosome by ectopically expressing ATG4B^C47A^, a mutant ATG4B that inhibits the lipidation of ATG8 family proteins, results in severe cytoplasmic vacuolation and cell death [418], suggesting that HCV may utilize autophagy to counteract cell death [418]. Notably, this study mentioned the differential impacts of HCV viral RNA replication (Con 1b replicon) and infectious HCVccon autophagy [418].

Apart from the replication of viral RNA and infectious HCVcc, several HCV viral proteins have also been shown to induce autophagy. Su and colleagues reported that HCV NS4B is the only viral protein that sufficiently induces incomplete autophagy via residues 1~190 (Table 1) [419]. Gregoire et al. showed that autophagy can be activated by ectopic expression of HCV NS3 alone (Table 1) [420]. In addition to NS4B and NS3, expression of the HCV core protein can induce complete autophagy (Table 1) [421,422] and another later study showed that HCV NS5A can trigger autophagy (Table 1) [423,424]. Although these results collectively suggest that HCV can activate autophagy via individual viral proteins, many conclusions were drawn based on the transient expression of viral proteins in cells that are not permissive to complete HCV life cycle, which may have led to large discrepant results.

The molecular mechanism by which HCV exploits to initiate autophagy has been investigated. Several groups have reported that HCV may activate ER stress, which is required for autophagy activation (Table 1) [401,405,406,409]. In addition to this mechanism, UPR inhibitors have been shown to suppress HCV viral RNA replication and viral-induced autophagy [438]. Wang et al. further reported that the HCV core protein triggers the UPR to upregulate DNA damage-inducible transcript 3 protein (DDIT3, also known as CHOP) expression, which consequentially upregulates the transcription of LC3B and ATG5 to activate autophagy (Table 1) [421]. In contrast to these studies, Bjorn-Patrick and colleagues reported that HCV infection activates autophagy independent of the UPR and that the UPR is not required for HCV growth [437].

HCV was also shown to induce ER stress to interfere with protein kinase B (PKB)-tuberous sclerosis (TSC)-mTOR complex 1 (mTORC1) signaling and thus activate autophagy (Table 1) [439]. Analogously, Shrivastava and colleagues also showed that HCV induces autophagy by elevating Beclin 1 expression and activating mTOR signaling (Table 1) [440]. Apart from ATGs functioned in autophagosome biogenesis, their physiological role in phagophore formation and autolysosome maturation in HCV-autophagy interactions has been studied. STX17, which functions in the autophagosome-lysosome fusion process, was first shown to control HCV egress by regulating the equilibrium between the release of a mature virion and the degradation of intracellularly retained viral particles within the lysosome (Table 1) [425]. On the other hand, Wang and colleagues showed that the STX17-mediated homotypic fusion of phagophores enhances HCV-induced autophagosome formation to benefit HCV viral RNA replication (Table 1) [441]. Very recently, alcohol was shown to enhance PIAS family protein (PIASy) expression to activate autophagy and thus promote HCV replication (Table 1) [426]. Recent studies implied that alternatively spliced forms of ATG10 can differentially modulate autophagic flux to regulate HCV replication (Table 1) [427,428].

In addition to acting as a pro-viral factor benefiting viral growth, autophagy has also been shown to eliminate LDs and mitochondria (Table 1) [429,430]. Vescovo et al. first showed that the expression of an autophagy marker, the lipidated LC3, is oppositely correlated with the clinical parameters related to steatosis in liver biopsies of chronic HCV-infected patients (Table 1) [429]. Their study further demonstrated that autophagy promotes the degradation of LDs in HCV replicon cells [429], suggesting that HCV-activated autophagy catabolizes LDs to circumvent the HCV-induced excess of LD accumulation in infected cells [429]. In addition, HCV was also shown to enhance mitophagosome formation to eliminate mitochondria in infected cells in a PINK-Parkin-dependent manner (Table 1) [430,442]. Moreover, this PINK-Parkin-mediated clearance of mitochondria by autophagy is required for replication of HCV viral RNA in infected cells [430]. Further study demonstrated that HCV triggers mitochondrial fission to promote the Parkin-mediated degradation of mitochondria, thus attenuating cell apoptosis and establishing viral persistence (Table 1) [431]. In contrast, Hara et al. reported that the HCV core may interact with Parkin to interfere with the translocation of Parkin into mitochondria, thus alleviating mitophagy and sustaining HCV-triggered mitochondrial damage (Table 1) [432]. Very recently, the replication of HCV viral RNA was implicated to trigger selective autophagy to engulf ubiquitinated aggregates within the viral-induced autophagic vacuoles (Table 1) [443]. On the other hand, HCV-induced CMA was shown to participate in the degradation of IFN-alpha receptor-1 (IFNAR1), which is stimulated by free fatty acid (FFA) (Table 1) [433]. Moreover, Matsui and colleagues recently reported that HCV NS5A can interact with Hsc70, a regulator of CMA, to target hepatocyte nuclear factor 1α (HNF1α) for lysosomal degradation (Table 1) [424]. A recent study implied that HCV-induced IFN-β-inducible SCOTIN recruits NS5A to autolysosomes for degradation, thus restricting HCV replication (Table 1) [434]. Together, these studies imply that HCV may utilize autophagy to counteract active viral replication and the host stress response by eliminating specific cargos.

Little is known about whether HCV indeed activates autophagy to regulate virus-host interactions in vivo, although increasing evidence implies a major pro-viral role of autophagy in the HCV replication cycle in infected cells of the in vitro hepatocyte cell culture model. The lack of a reliable small animal system for investigating the HCV life cycle and host cellular response and the inconvenient monitoring of HCV-host interactions in HCV-infected liver specimens account for the slow progress in elucidating the detailed physiological role of autophagy in the HCV life cycle in HCV-infected cells. Notably, whether autophagy is activated in host cells infected by different HCV genotypes is still unresolved because the only infectious HCVcc used in the in vitro model, HCV JFH1 (genotype 2a), harbors an extraordinary replication cycle that is quite different from that of the prevalent genotypes 1 and 3 in most infected individuals [230,402].

In addition to promoting HCV growth, viral-triggered autophagy seems to degrade a panel of intracellular components, including LDs, mitochondria and host and viral proteins [424,429,430,434,443]. However, these conclusions are largely debated to the previous study shown that HCV activates incomplete autophagy by inhibiting autophagic flux [402,418]. Given that both HCV replication and autophagy require enormous rearrangement of the host intracellular membrane, whether these proteins and organelles are selectively engulfed into HCV-activated autophagic machinery before the specific cargo-degrading receptors are identified remains unknown. Moreover, whether the elimination of these cargos participates in the development of HCV-associated liver diseases also requires further elucidation.

Autophagy has emerged as a critical player in balancing lipid metabolism by the selective degradation of LDs, that is, lipophagy [163] and by regulating LD biogenesis [444,445]. Moreover, further study showed that autophagy controls the accumulation of bodily lipids and regulates adipocyte differentiation [446]. Together, these results indicate that alteration of autophagy may interfere with lipid catabolism and metabolic homeostasis. Furthermore, autophagy has been negatively correlated with the clinical parameters of liver steatosis in liver specimens from HCV-infected patients [429] and HCV-activated autophagy has been shown to degrade mitochondria [430]. Therefore, further study is needed to examine whether HCV-activated autophagy alters metabolic homeostasis and organelle regeneration, thus accelerating the progression of HCV-related metabolic diseases. On the other hand, autophagy has been demonstrated to be critical for the suppression of tumor development. The gene knockout of ATGs, such as Beclin 1 and ATG5, in mice has been shown to interfere with the autophagic process and promote the development of malignant tumors [447,448,449,450,451]. Furthermore, several studies have shown high p62/SQSTM1 expression in various types of HCC tissues and cell lines [450,452], suggesting that autophagy interference is a major route for the induction of hepatocarcinogenesis. HCV infection likely initiates an autophagic process to benefit viral growth, perhaps repressing basal autophagy to alter metabolic homeostasis and induce the development of liver cancer. Accordingly, a recent study implied that p62/SQSTM1 is highly phosphorylated at serine 349 in the tumor tissues of HCV-positive HCC patients and that phosphorylated p62/SQSTM1 drives the glucoronate and glutathione synthesis pathways to enhance drug resistance and malignancy of liver cancer [453]. Increasing evidence has demonstrated that tumor cells may trigger autophagy to circumvent the tumor microenvironment and thus increase cell proliferation potency [454,455,456,457,458,459,460,461] and counteract a variety of stress responses, such as hypoxia and nutrient deprivation [457,458,461]. Moreover, autophagy has also been shown to induce drug resistance in anticancer treatment [454,455,456]. Together, these studies suggest that HCV infection may activate autophagy to establish a cell microenvironment that favors cell surveillance under stress conditions in chronically infected patients, possibly promoting the development of HCV-associated end-stage liver diseases. Nevertheless, a feasible and small animal model that enables investigation of the complete HCV life cycle in the liver is urgently needed to uncover the impact of HCV-autophagy interactions on the progression of HCV-related liver diseases.

### 4.2. DENV

Analogous to HCV, DENV infection also activates host autophagy in infected cells [398,462,463,464,465]. Lee and colleagues first showed that DENV serotype 2 (DENV-2) infection induces autophagy in infected Huh7 cells, as demonstrated by the formation of autophagosomes and the increased level of LC3 lipidation (Table 2) [398]. The fact that genetically silencing ATG5 by RNA interference inhibited DENV replication implies the pro-viral effect of host autophagy on promoting DENV replication [398]. Soon afterward, Panyasrivanit et al. also reported that DENV-2 induces autophagy in viral-infected HepG2 cells (Table 2) [465]. The colocalization of dsRNA, DENV NS1 and ribosomal protein L28 within LC3-labeled autophagic vacuoles and the observation of mannose-6-phosphate receptor (MPR)-enriched amphisomes suggest that DENV-induced autophagy machinery functions as a platform for DENV replication [465,466]. Interference of autophagy initiation by 3-MA dramatically reduces the intracellular amount of viral RNA and extracellular virions [465,466], implying that the autophagic process is required for organization of the DENV replication compartment. In contrast, inhibition of autophagosome fusion with lysosomes by L-Asparagine (L-Asn) slightly increases the levels of intracellular viral RNA and extracellular virions [465,466], suggesting the potential role of autolysosomes in eliminating DENV. In addition to DENV-2, DENV-3 was also shown to induce autophagy to form autolysosomes that encompass cathepsin D, NS1 and dsRNA and support viral RNA replication in HepG2 cells (Table 2) [464], again arguing that DENV infection activates autophagy to reconstitute membranous structures for viral growth. In contrast, DENV-2 infection of U937 cells induces the accumulation of autophagosomes by interfering with their fusion with lysosomes (Table 2) [462]. Interruption of autophagy initiation by a dominant Vps34/PI3KC3 mutant decreases the replication of DENV viral RNA and the secretion of infectious DENV particles, indicating that DENV-2 infection also promotes autophagy to benefit the viral life cycle. Additionally, this study implies that DENV infection differentially modulates autophagy in different infectious cell models. The induction of autophagy by DENV infection was further confirmed by an in vivo study on DENV infection in mice (Table 2) [467]. In this study, Lee and colleagues demonstrated that DENV infection triggers the formation of autophagic vacuoles and the lipidation of LC3 in infected brain tissues (Table 2) [467]. Most importantly, the inhibition of autophagy in DENV-infected mice represses viral replication and the forwarded induction of DENV-induced autophagy by rapamycin promotes disease progression, supporting a new idea in which autophagy not only plays a pro-viral role in DENV growth but also participates in the pathogenesis of DENV-associated diseases.

Apart from viral RNA replication within autophagic machinery, the pharmacological modulation of autophagy in DENV-infected cells showed that DENV infection activates autophagy to facilitate the maturation of infectious particles (Table 2) [468]. Notably, autophagy was shown to facilitate DENV infection via the ADE-mediated pathway (Table 2) [469] and to associate with DENV infectious particles to escape antibody neutralization and promote cell-to-cell transmission [470]. Moreover, Chu et al. reported that autophagy interacts with DENV during the initial stage of infection to facilitate the DENV entry process (Table 2) [471]. Furthermore, Bartenschlager’s group demonstrated that DENV infection enhances autophagic flux to promote viral replication during the initial stage of infection and induces the degradation of p62/SQSTM1 via a proteasomal mechanism (Table 2) [472]. Together, these studies imply multiple roles of autophagy in the DENV life cycle.

In addition to playing a pro-viral role in the DENV viral life cycle, autophagy activated by DENV infection also alters cellular metabolism, as first demonstrated by Heaton et al. (Table 2) [463]. This study demonstrated that DENV triggers the autophagic process all the way through autolysosome maturation in various types of cells to catabolize LDs to release FFA and generate ATP for the replication of DENV [463]. Very recently, Zhang et al. reported that DENV NS4A interacts with ancient ubiquitous proteins 1 (AUP1), an LD-associated VLDL assembly regulator, to promote DENV-induced lipophagy, thus promoting virus production (Table 2) [473]. Another study demonstrated that DENV infection activates the AMPK-mTOR axis to promote lipophagy (Table 2) [474]. These results collectively conclude that DENV activates autophagy to promote LD catabolism, thus increasing the β-oxidation of FFA for viral replication.

Several studies have investigated how DENV infection induces autophagy activation and DENV NS4A has been shown to activate autophagy in a PI3K-dependent manner (Table 2) [475]. Additionally, NS1 and NS4B reportedly participate in DENV-induced autophagy, promoting the degradation of LDs and viral-induced vascular leakage (Table 2) [473,476]. In a similar fashion to HCV, DENV also triggers the ER stress/UPR pathway to activate autophagy, thus promoting viral replication and cell surveillance (Table 2) [477,478,479].

Although autophagy was extensively shown to be activated by DENV infection, little is known about how DENV initiates this process. Additionally, the exact physiological significance of viral-induced autophagy in DENV-host interactions and the pathogenesis of DENV-associated diseases remain under further investigation. Comprehensively understanding the functional roles of DENV-activated autophagy in host cells will not only help unveil the progression of DENV-related diseases but also provide a molecular basis for the rational design of antiviral treatment.

### 4.3. JEV

JEV infection was first reported to activate autophagy in human NT-2 cells (a pluripotent human testicular embryonal carcinoma cell line), as demonstrated by increased levels of lipidated LC3 (Table 3) [480]. This study also showed that induction of autophagy by rapamycin enhances JEV replication, whereas the autophagy inhibitor 3-MA suppresses viral growth [480]. Silencing of ATG5 and Beclin 1 genes in JEV-infected cells reduces viral replication, suggesting that JEV activates autophagy to benefit viral growth [480]. The lack of colocalization between dsRNA and LC3-labeled autophagic vacuoles suggests that JEV-induced autophagy machinery does not provide membrane compartments for JEV replication [480]. Moreover, the specific colocalization of LC3-labeled autophagic vacuoles with JEV during the initial stage of infection implies that viral-activated autophagy participates in the JEV entry process. Later, Jin et al. also demonstrated that JEV infection activates the entire autophagic process through autolysosome maturation, which is required for JEV replication (Table 3) [481]. Interference with autophagy by knocking down the ATG5 and Beclin 1 genes abolishes the replication of JEV viral RNA, accompanied by enhanced cell apoptosis and induced antiviral immunity [481]. On the other hand, the loss of p62/SQSTM1 in mouse embryonic fibroblasts downregulates the viral replication of JEV (Table 3) [482], suggesting that the p62/SQSTM1-mediated autophagic process promotes JEV growth in infected cells. Moreover, the C, PrM and NS3 proteins were shown to induce JEV-induced autophagy, presumably via IRGM (Table 3) [483]. Similar to HCV and DENV, the ER stress/UPR pathway is required for JEV-induced autophagy activation in neuronal cells (Table 3) [484]. A recent study reported that diphenyleneiodonium, an antioxidant drug, represses JEV-activated autophagy by inhibiting the ER/UPR pathway to suppress JEV production in neuronal Neuro2a cells (Table 3) [485], highlighting that interfering with JEV-activated UPR and autophagy might be a therapeutic target for inhibiting JEV infection. In contrast, JEV infection was also shown to attenuate autophagy to promote viral replication via the ubiquitin E3 ligase activity of Nedd4 and organization of the nonlipidated LC3-containing EDEM1 (ER degradation enhancer, mannosidase α-like 1)-associated membrane (Table 3) [486,487]. Thus far, little is known about whether JEV-activated autophagy participates in the progression of neurological diseases but this information could be obtained by investigating how JEV induces autophagy and comprehensively understanding the detailed functions of autophagy in the JEV viral life cycle.

### 4.4. WNV

In contrast to the pro-viral role of autophagy in DENV and JEV, autophagy is not likely required for activation of the WNV viral life cycle. WNV infection was shown to induce LC3 lipidation and the formation of LC3-labeled autophagic vacuoles in Vero cells (Table 4) [488]. The loss of ATG5 and related autophagy by knockdown and knockout methods, respectively, did not alter WNV replication in infected cells [488], suggesting that WNV-activated autophagy is not required for the infectious life cycle of WNV. This conclusion is further supported by other studies showing that depletion of ATG5 and ATG7 has no significant effect on the production of WNV infectious particles (Table 4) [489,490] and the supporting role of mTOR signaling for the translation of WNV viral RNA (Table 4) [491]. These studies provided a mechanistic basis for developing an autophagy inducer, Tat-Beclin 1, to inhibit WNV replication and disease progression in WNV-infected mice (Table 4) [490,492]. Unlike DENV and JEV, the UPR is not coupled or required for the activation of WNV-induced autophagy [493]. Collectively, these studies suggest that autophagy may play an antiviral role in restricting WNV infection. However, whether autophagy participates in the pathogenesis of WNV-associated neurological diseases remains unknown and successfully answering this question will facilitate the design of a feasible and effective antiviral therapy against WNV infection.

### 4.5. ZIKV

In recent years, Zika was discovered to activate autophagy to induce the formation of autophagosomes and thus promote viral replication in human skin cells (Table 5) [494]. Additionally, the modulation of autophagy by pharmacological regulators is associated with ZIKV replication in infected cells [494], supporting the notion that autophagy plays a pro-viral role in ZIKV infection. Moreover, ZIKV NS4A and NS4B were demonstrated to induce autophagy in human fetal neural stem cells (fNSCs) by deregulating Akt-mTOR signaling (Table 5) [495]. This Zika NS4A- and NS4B-mediated autophagy activation is required for ZIKV replication and the impaired neurogenesis of fNSCs [495]. Very recently, ZIKV infection of human umbilical vein endothelial cells (HUVECs) was shown to induce complete autophagy and thus promote p62/SQSTM1 degradation, which is critically required for ZIKV replication (Table 5) [496]. These studies indicated that ZIKV-activated autophagy may participate in the pathogenesis of ZIKV-related diseases and suggest that the inhibition of ZIKV-induced autophagy may represent a therapeutic target for inhibiting ZIKV infection. This hypothesis is further supported by a recent study showing that deletion of ATG16L and treatment with chloroquine, an inhibitor of autolysosome maturation, restricts ZIKV vertical transmission in pregnant mice and ameliorates the placental and fetal outcomes of ZIKV infection (Table 5) [497]. However, recent studies indicated that inflammation-induced autophagy limits ZIKV infection via the stimulator of interferon genes (STING) in the Drosophila brain (Table 5) [498], implying that ZIKV-activated autophagy may represent a new route for eliminating ZIKV infection via inflammation. The physiological significance of autophagy in the life cycle and transmission of ZIKV as well as the mechanisms underlying the pathogenesis of ZIKV-induced neonatal disorders continue to be elucidated and this information is urgently required for development of an effective anti-ZIKV drug for clinical use. 

## 5. Autophagy as a Therapeutic Target for Curing Flaviviruses Infection

Suppression of autophagy has been demonstrated to inhibit viral growth in most of flavivirus-infected cells. Interference with membrane nucleation by the Vps34/PI3KC3 inhibitor, 3-MA reduces the viral replications of DENV [398,462,463,464,465,466,467,471], JEV [480,483], WNV [488] and ZIKV [497]. It is noted that 3-MA increases the survival rate and mitigates the symptoms of DENV-infected mice [467,468], implying that autophagy inhibition can be used to control disease progress in DENV infection in vivo. Inhibition of maturation and activity of autolysosomes by chloroquine and bafilomycin-A1 also has an inhibitory effect on virus growth in HCV [497,499], DENV [500,501,502,503], JEV [504,505] and ZIKV [495,496,497,506,507,508]. In addition, chloroquine was also shown to mitigate DENV-related symptoms in patients [509,510,511] and inhibit the maternal-fetal transmission of ZIKV and protect fetal mice from microcephaly caused by ZIKV infection [497,507,508]. Moreover, spautin-1 was also shown to reduce the maturation of DENV virions [468]. These studies collectively indicate that the inhibition of host autophagy may present a feasible strategy of restricting flavivirus infection. However, deregulated autophagy in some cases of flaviviral infection leads to the cell death of infected cells, such as HCV [407,418] and JEV [481,487], suggesting a risk of unexpected tissue damage by viral infection, which could be caused by a defect in the cell survival function of autophagy. In contrast, induction of autophagy by Tat-Beclin 1 suppress WNV infection in vitro and in vivo [490,492], implying that induction of autophagy may promote the xenophagic elimination of infectious WNV. Although modulation of autophagy, particularly the inhibition of autophagy can be conceivably utilized to combat flaviviral infection, the specific autophagy inhibitors that contain higher potency and lower cytotoxicity still remain to be discovered. Given that the roles of autophagy in flavivirus infection is context- and disease-dependent, the impacts of autophagy inhibitors in the cytoprotective and pro-survival functions of autophagy should be investigated before the use of these strategies in clinical medicine [5,6,512,513]. Interference with host autophagy may provide for virus infection control but may also deregulate metabolism homeostasis and disrupt the elimination of damaged materials caused by viral infection in host cells. For instance, the inhibition of HCV infection by the repression of autophagy may introduce the risk of developing liver-related diseases since autophagy defects have been shown to induce several types of diseases in liver [514,515]. Before we attempt to inhibit autophagy for therapeutic intervention in flavivirus infection, a deeper and more comprehensive understanding of flaviviral viruses and host interactions is needed. In addition, the different types of treatment strategies that are disease progress dependent and the improved safety of drugs for clinic use are urgently required. 

## 6. Conclusions and Future Directions

Autophagy plays a pivotal role in balancing cellular homeostasis via the continuous degradation of intracellular components and recycling of nutrients for biogenesis. Upon the infection of cells with flavivirus, autophagy is exploited by the virus to reconstitute the membrane structure required for completion of the viral life cycle and degrade harmful organelles in the infected cells. Additionally, autophagy is activated to allow flaviviruses to escape innate immune surveillance and protect infected cells from death. Although the interactions between flaviviruses and host autophagy have been extensively studied over the past decade, several fundamental questions regarding how flaviviruses activate autophagy and how viral-induced autophagy influences the pathogeneses of flavivirus-related diseases remain unknown. Future investigations are needed to understand the clinical relevance of autophagy and flavivirus infections in infected patients. For instance, whether the conclusions drawn from these in vitro infected cell models can be recapitalized in real in vivo physiological conditions needs to be determined. Additionally, the lack of a feasible and conceivable mouse model for most flaviviruses substantially hampers the development of a dynamic and timely assay for assessing flavivirus-induced autophagy in vivo. Dissecting the interplay between flaviviruses and autophagy in an in vivo animal that supports the complete life cycle will permit the detailed elucidation of the functional roles of autophagy in viral infection, host defense regulation and disease progression. An innovative high-throughput screening strategy and bioinformatics analysis tools will improve the comprehensive understanding of autophagy-flavivirus interactions and help identify potential autophagy therapeutic targets to design an effective antiviral strategy for curing flavivirus infection in the clinic.

## Figures and Tables

**Figure 1 ijms-19-03940-f001:**
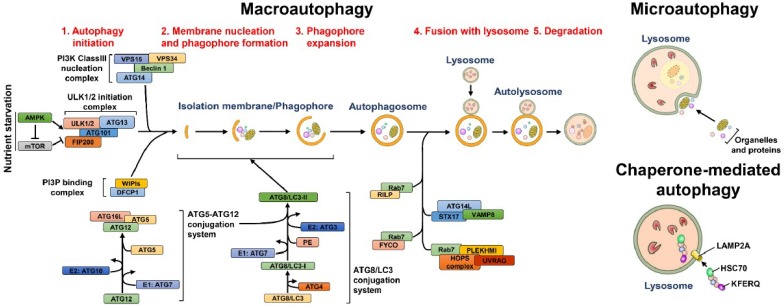
Overview of autophagy. Three types of autophagy, including macroautophagy, microautophagy and chaperone-mediated autophagy, have been identified. Macroautophagy undergoes a stepwise vacuole biogenesis process that sequestrates the intracellular components within autophagosomes, which finally fuses with lysosomes to degrade the engulfed cargoes. Microautophagy is an invagination and scission process of lysosomal membrane that directly engulfs the intracellular portions to the lumen of lysosomes for degradation. Chaperone-mediated autophagy (CMA) involves the recognition of substrates that contain KFERQ motifs via the heat shock cognate protein of 70 KDa (Hsc70) and their delivery into the lysosomal lumen through the lysosomal membrane protein 2A (LAMP2A). For macroautophagy, two kinds of metabolic sensors, the mammalian target of rapamycin (mTOR) complex and 5′-AMP-activated protein kinase (AMPK) differentially regulate autophagy initiation. When cells are starved of nutrients, AMPK act as a positive regulator to activate autophagy through inhibiting mTOR. The inhibition of mTOR leads to translocation of unc-51 like-kinase (ULK) complex (ULK1/2, ATG13, RB1-inducible coiled-coil 1 (RB1CC1, also known as FIP200) and ATG101) to autophagy initiation site. Then, the ULK complex recruits and activates the class III phosphatidylinositol-3-OH kinase (class III-PI3K complex, including Vps34/PI3KC3, Vps15, Beclin 1 and ATG14) to generate PtdIn(3)P. The newly synthesized PtdIn(3)P recruits the double-FYVE-containing protein 1 (DFCP1) and WD-repeat domain PtdIns(3)P-interacting (WIPI) family proteins to form the isolation membrane (IM)/phagophore. Two ubiquitin-like (UBL) conjugation systems are required for the expansion and elongation of phagophore to form autophagosomes. The ubiquitin conjugation enzyme 1 (E1) ATG7 activates ATG12 via the formation of a thioester bond between the C-terminal glycine of ATG12 and the cysteine residue of ATG7. Then ATG12 is transferred to ATG10 (E2) and subsequently conjugated to ATG5, yielding an ATG5-ATG12 complex. Finally, the ATG12-ATG5 conjugate interacts with ATG16L to form an ATG12-ATG5-ATG16L complex. ATG8/LC3 family proteins are cleaved by a cysteine protease ATG4 to generate the ATG8/LC3-I. Then ATG8/LC3-I is covalently linked to phosphatidylethanolamine (PE) to form the lipidated form of LC3 (ATG8/LC3-II) through enzymatic reactions of the ATG7 E1 and ATG3 E2. The mature autophagosomes fuse with lysosomes to form autolysosomes, in which the sequestrated materials are degraded. The small GTPase Ras-related protein 7 (Rab7) regulates the fusion of autophagosomes with lysosomesby interacting with cytoskeleton-associated factors, the FYVE and coiled-coil domain-containing 1 (FYCO1) and Rab-interacting lysosomal protein (RILP). Moreover, the concerted actions of multiple proteins on the HOPS complex, including sytaxin17 (STX17), the UV radiation resistance-associated (UVRAG), ATG14 and the pleckstrin homology domain-containing protein family member 1 (PLEKHM1) also participate in the maturation process of autolysosome.

**Figure 2 ijms-19-03940-f002:**
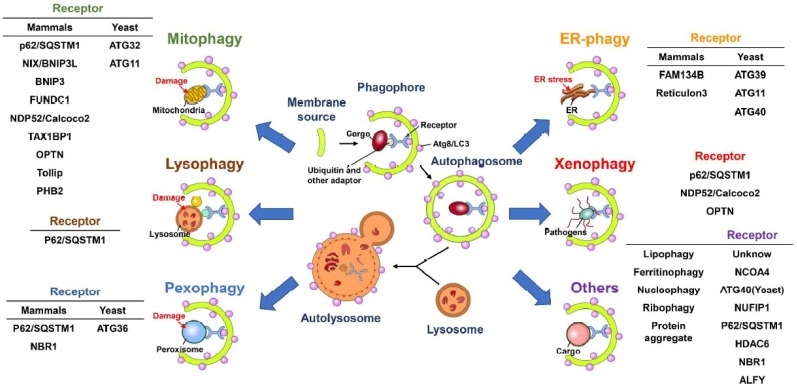
Selective autophagy and cargo receptors. Selective autophagy involves the targeting of cargos to autophagic process via the receptor proteins that contains an ATG8/LC3-interacting regions (LIRs) for the binding to ATG8/LC3 located on the membrane of IM/phagophore. The autophagosomes elongated from IM/phagophore fuse with lysosomes to form autolysosomes, in which the engulfed cargos are degraded. The ubiquitination of cargo or an additional adaptor protein is often required for the recognition process between cargos and receptors proteins. Selective autophagy participates in the elimination of various kinds of organelles and proteins. Mammalian and yeast cargo receptors responsible for the degradation of the corresponding cargos by selective autophagy are listed as indicated. Selective autophagy participates in the degradations of damaged mitochondria, injured lysosomes, damaged peroxisomes, stressed endoplasmic reticulum (ER) and infected pathogens within autophagic degradation (mitophagy, lysophagy, pexophagy, ER-phagy and xenophagy, respectively). Moreover, other cargos, including lipid droplets (LDs) ferritin, nuclei, ribosomes, protein aggregates can be selectively sequestrated by selective autophagy for degradation via different types of cargo receptors.

**Table 1 ijms-19-03940-t001:** Summary of HCV-autophagy interactions.

Genotype /Serotype	Experimental Model	Characteristics of Autophagy	Functional Target	Reference
H77 (1a)	Immortalized human hepatocytes (IHH)/Infection Human hepatocellular carcinoma, Huh7.5 cells /Infection	Transmission electron microscopy observation of autophagic vacuoles in the infected cells Immunofluorescence detection of GFP-LC3-labeled punctate structure in the infected cells Upregulations of ATG5-ATG12 conjugate and Beclin in the infected cells	Unknown	Ait-Goughoulte et al. [399]
JFH1 (2a)	Human hepatocellular carcinoma, Huh7.5 cells/Viral RNA transfection	Induction of LC3-I to LC3-II conversion in the viral RNA-transfected cells Increased formation of GFP-LC3-labeled-autophagosomes that are not colocalized with lysosomes in the viral RNA-transfected cells (Incomplete autophagy) Lack of enhancement of degradation of long-lived proteins in the viral RNA-transfected cells Activation of autophagy by unfolded protein response (UPR) in the viral RNA-transfected cells Induction of UPR by transfection of viral RNA Inhibited the replication of viral RNA by interference with UPR-mediated autophagy	Promotion on viral RNA replication	Sir et al. [401]
JFH1 (2a)	Human hepatocellular carcinoma, Huh7 cells/Infection	Induction of LC3-I to LC3-II conversion in the infected cells Increased formation of GFP-LC3-labeled-autophagosomes in the infected cells Enhanced the translation of incoming viral RNA in the infected cells Reduced the intracellular viral RNA level and extracellular amount of virion in the infected cells by knockdown of ATG4B and Beclin 1 No significant effect on viral replication of established infection by knockdown of ATG4B and Beclin 1 No apparent colocalization between NS proteins and autophagic vacuoles in infected cells	Promotion on viral RNA replication Support on the translation of viral RNA	Dreux et al. [405]
JFH1 (2a)	Human hepatocellular carcinoma, Huh7 cells/Infection	Increased conversion of LC3-I to LC3-II in the infected cells Transmission electron microscopy observation of early- and late-staged autophagic vacuoles in the infected cells Immunofluorescence detection of GFP-LC3-labeled punctate structure in the infected cells Enhanced autophagic flux by virus infection Activation of autophagy by UPR in the infected cells Reduced viral RNA replication in infected cells by knockdown of UPR and autophagy genes Induction of HCV pathogen-associated molecular pattern-mediated interferon-(IFN-β) by knockdown of UPR and autophagy genes	Promotion on viral RNA replication Suppression of innate antiviral immunity	Ke and Chen [406,410]
JFH1 (2a)	Immortalized human hepatocytes (IHH)/Infection Human hepatocellular carcinoma, Huh7.5 cells /Infection	Reduced viral growth in the infected cells by knockdown of Beclin 1 Induction of autophagosome fusion with lysosome in the infected cells Increased levels of IFN-β and IFN-stimulated genes (ISGs) in the infected cells by knockdown of Beclin 1and ATG7 Promoted cell apoptosis of infected cells by knockdown of Beclin 1and ATG7	Promotion on viral RNA replication Suppression of innate antiviral immunity Protection the infected cells from cell death	Shrivastava et al. [407]
JFH1 (2a)	Human hepatocellular carcinoma, Huh7 cells/Infection	Degradation of TRAF6 by autophagy in the infected cells Inhibited TRAF6 degradation in the infected cells by bafilomycin A1 (autophagy inhibitor) Colocalization of TRAF6 with autophagic vacuoles in the infected cells Promoted TRAF6 degradation via p62-dependent autophagy in the infected cells Increased virus replication in the infected cells by knockdown of TRAF6 Reduced NF-kB signaling response by knockdown of TRAF6	Promotion on viral RNA replication Suppression of innate antiviral immunity	Chan et al. [408]
JFH1 (2a)	Human hepatocellular carcinoma, Huh7 and Huh7.5.1 cells/Infection and transfection of replicon RNA	Induced LC3-I to LC3-II conversion by transfection of replicon RNA Reduced of replication of replicon viral RNA in replicon cells by knockdown of LC3 and ATG7 Colocalization of NS5A and NS5B with autophagosome in infected cells Colocalization of viral RNA with autophagosome in infected cells Co-immunoprecipitation of the replication complex with autophagosomes in infected cells	Promotion on viral RNA replication Support on the organization of replication complex for viral RNA	Sir et al. [411]
JFH1 (2a)	Human hepatocellular carcinoma, Huh7 cells/Infection	Transient interactions between ATG5 with NS5B and NS4B at the initial infecting stage Inhibited viral replication in infected cells by knockdown of ATG5	Promotion on viral RNA replication Support on the organization of replication complex for viral RNA	Guevin et al. [412]
JFH1 (2a)	Human hepatocellular carcinoma, Huh7.5.1cells/Transfection of replicon RNA	Electron micrograph of autophagosome in the replicon cells Colocalization of NS proteins, viral RNA and LC3 with double-membraned vesicles (DMVs) in replicon cells Induced formation of DMVs by virus-induced autophagy in replicon cells	Promotion on viral RNA replication Support on the organization of replication complex for viral RNA	Ferraris et al. [413]
JFH1 (2a)	Human hepatocellular carcinoma, Huh7 and Huh7.5.1 cells/Infection and transfection of replicon RNA	Increased GFP-LC3-labeled autophagic vacuoles in the replicon RNA-transfected cells Co-fractionation of NS5A with purified autophagosomes in the replicon RNA-transfected cells Colocalization of GFP-LC3-labeled autophagic vacuoles with the components of lipid raft in replicon cells Colocalization of GFP-LC3-labeled autophagic vacuoles with caveolin 1 and NS5A in replicon cells	Promotion on viral RNA replication Support on the recruitment of lipid rafts for viral RNA replication	Kim et al. [414]
JFH1 (2a)	Human hepatocellular carcinoma, Huh7.5.1 cells/Infection	Induced formation of GFP-LC3-labeled punctate structure No significant colocalization between NS proteins and autophagic vacuoles Impaired virion release in the infected cells by knockdown of Beclin 1 and ATG7	Promotion on the release of viral particles	Tanida et al. [415]
H77 (1a) JFH1 (2a)	Immortalized human hepatocytes (IHH)/Infection Human hepatocellular carcinoma, Huh7.5 cells /Infection	Promoted autophagosome fusion with lysosome by virus infection Accumulated intracellular virion in the infected cells by knockdown of Beclin 1 and ATG7 Reduced intracellular virion in the infected cells by knockdown of Beclin 1 and ATG7 Induced accumulation of exosome in the infected cells by knockdown of Beclin 1 Inhibited release of exosome-associated virion in the infected cells by knockdown of Beclin 1	Promotion on the release of viral particles	Shrivastava et al. [416]
JFH1 (2a)	Human hepatocellular carcinoma, Huh7 and Huh7.5.1 cells /Infection and transfection of replicon RNA	Colocalization of apolipoprotein A (ApoE) with GFP-LC3-labeled autophagosomes in replicon cells Colocalization of ApoE with GFP-LC3-labeled autophagosomes in the infected cells Degradation of ApoE by autophagy in the infected cells Inhibited ApoE degradation by autophagy inhibition in replicon and infected cells Reduced the extracellular amount of viral particles in the infected cells by knockdown of ATG7 Colocalization of ApoE and E2 protein in the infected cells	Promotion on the release of viral particles	Kim et al. [417]
JFH1 (2a) Con1 (1b)	Human hepatocellular carcinoma, Huh7 cells/Transfection of replicon RNA	Induction of autophagy in replicon cells Impaired autophagic flux in replicon cells Inhibited autophagy maturation in replicon cells Enhanced secretion of pro-cathepsin B in replicon cells Induction of sever cytoplasmic vacuolation by inhibition of autophagosome formation	Counteracting the viral-induced cell death	Taguwa et al. [418]
JC1 (2a)	Human hepatocellular carcinoma, Huh7.5 cells/Infection and transfection of viral proteins	Induction of autophagy by NS4B transfection Induced formation of autophagic vacuoles by NS4B protein Involvement of Rab5 and PI3K/Vps34 in the activation of autophagy by NS4B and virus infection Interactions of NS4B with Rab5, PI3K/Vps34 and Beclin 1	Unknown	Su et al. [419]
JFH1 (2a)	Human hepatocellular carcinoma, Huh7.5 cells/Infection	Activation of autophagy through immunity-associated GTPase family M (IRGM) in the infected cells Modulation of virus production in the infected cells by IRGM Induction of autophagy in the infected cells by interaction between NS3 and IRGM	Promotion on viral RNA replication	Gregoire et al. [420]
JFH1 (2a)	Human hepatocellular carcinoma, Huh7 cells/ Transfection of viral proteins	Induction of autophagy by expression of core protein Enhanced autophagic flux by core proteins Activation of complete autophagy by core protein Induction of ER stress and UPR by core protein Activation of autophagy by UPR in core-expressing cells Transcriptional activation of ATG12 and LC3 through UPR-DDIT3 signaling in core-transfected cells	Unknown	Wang et al. [421] Liu et al. [422]
Genotype 1b	Human hepatocellular carcinoma, HepG2 cells/Transfection of viral proteins Human hepatic cell line L02/Transfection of viral proteins	Induction of autophagy by expression of NS5A protein Activation of autophagy through NS5ATP9 in NS5A-transfected cells Induction of NS5ATP9-mediated autophagy via transcriptionally activation of Beclin 1 in NS5A-transfected cells	Unknown	Quan et al. [423]
J6/JFH1 (2a)	Human hepatocellular carcinoma, Huh7 cells/Infection and transfection of viral proteins	Interaction between the hepatocyte nuclear factor 1alpha (HNF1α) and Hsc70, a regulator of chaperone-mediated autophagy (CMA) Enhanced interaction between HNF1α and Hsc70 by NS5A Promoted HNF1α degradation by virus-induced autophagy Inhibited the CMA-mediated HNF1a degradation by knockdown of lysosomal associated protein 2A (LAMP2A) and Hsc70	Promotion on HNF1α degradation	Matsui et al. [424]
JFH1 (2a) J6 (2a)	Human hepatocellular carcinoma, Huh7 and Huh7.5.1 cells/Transfection of viral RNA	Reduced syntaxin 17 (STX17) expression in viral RNA-transfected cells Increased STX17 degradation by autophagy in viral RNA-transfected cells Downregulated virus production in viral RNA-transfected cells by overexpression of STX17 Enhanced the release of infectious particles in viral RNA-transfected cells by silencing of STX17 Promoted virus release of the viral RNA-transfected cells by STX17 degradation-mediated block of autolysosome formation	Facilitation on virion release	Ren et al. [425]
JFH1 (2a)	Human hepatocellular carcinoma, Huh7.5.1 cells/Infection and transfection of replicon RNA	Electron micrographic detection of homotypic fusion of phagophore in the infected cells Inhibited homotypic fusion of phagophore in the infected cells by knockdown of STX17 Requirement of STX17 for autophagosome formation in the infected cells Promotion of viral replication by organizing replication complex within phagophore	Promotion on viral RNA replication Support on the organization of replication complex for viral RNA	Wang et al. [424]
JFH1 (2a)	Human hepatocellular carcinoma, Huh7 cells/Infection	Enhanced virus replication in the infected cells by alcohol Activation of autophagy by alcohol Increased autophagic flux by alcohol Induction of PIAS family protein (PIASy) expression by alcohol Activation of autophagy by PIASy Promotion of virus replication by alcohol-induced PIASy and mediated autophagy	Promotion on viral RNA replication by alcohol-induced autophagy	Ran et al. [426]
J4L6s (1b)	Human hepatocellular carcinoma, HepG2 cells/Transfection of replicon RNA	Different regulation of viral RNA replication by alternatively spliced forms of ATG10 Differential activation of autophagic flux by alternatively spliced forms of ATG10 Differential modulation of innate immunity by alternatively spliced forms of ATG10 Interaction of the short form of ATG10 with interleukin 28 within autolysosome	Promotion on the degradation of replicon RNA	Zhao et al. and Zhang et al. [427,428]
1. Con1 (1b) 2. JFH1 (1a)	Human hepatocellular carcinoma, Huh7 cells/Infection and transfection of replicon RNA The liver biopsies of HCV-infected patients	Inverse relationship between LC3-I to LC3-II conversion and clinical parameters of steatosis Colocalization of RFP-LC3-labeled autophagic vacuoles with lipid droplets (LDs) in replicon cells Induction of cholesterol-targeting autophagy in the infected cells Increased cholesterol deposits by autophagy inhibition in replicon cells	Promotion on LDs catabolism	Vescovo et al. [429]
1. JFH1 (2a) 2. BM4–5 Feo (1b)	Human hepatocellular carcinoma, Huh7.5.1 cells /Infection and transfection of replicon RNA	Induction of mitochondrial damage by virus infection Induced translocation of Parkin into mitochondria in the infected cells Triggered ubiquitination of Parkin, mitochondrial proteins and p62/SQSTM1 Stimulated Parkin and PTEN-induced kinase 1 (PINK1) expressions Induction of complete mitophagosome Repressed viral replication by knockdown of Parkin and PINK1 Attenuated cell apoptosis and established viral persistence by mitophagosome in the infected cells	Promotion on mitochondria degradation Protection of infected cells from apoptosis Establishment of viral persistence	Kim et al. [430,431]
JFH1 (2a)	Human hepatocellular carcinoma, Huh7 cells /Infection and transfection of viral protein	Impaired translocation of Parkin to mitochondria in the infected cells Interaction between Parkin and core protein Suppressed the ubiquitination of mitochondrial proteins in the infected cells Repressed the formation of mitophagosome in the infected cells	Sustained mitochondrial injury	Hara et al. [432]
1. JFH1 (2a) 2. Con1 (1b)	Human hepatocellular carcinoma, Huh7 and Huh7.5.1 cells /Infection and transfection of replicon RNA	Activation of autophagy by virus infection and transfection of replicon RNA Colocalization of polyubiquitination and autophagic vacuoles in the infected cells Association between the polyubiquitination foci with replication complex and autophagic vacuoles	Unknown	Mori et al. [424]
JFH1 (2a)	Human hepatocellular carcinoma, Huh7 and Huh7.5.1 cells /Infection	Increased virus replication by free fatty acids (FFAs) in the infected cells Enhanced LDs formation in the infected cells by FFAs Block of innate antiviral immunity in the infected cells by FFAs Induced degradation of IFN receptor A1 (IFNAR1) by FFAs-induced CMA Interactions of IFNAR1 with the components of CMA	Suppression of innate antiviral immunity	Kurt el al. [433]
JFH1 (2a)	Human hepatocellular carcinoma, Huh7 cells/Infection	Inhibited virus replication in the infected cells by IFN-β-inducible SCOTIN Recruitment of NS5A protein to autophagosomes by SCOTIN Restricted virus infection by SCOTIN-mediated degradation of NS5A in autolysosome Degradation of SCOTIN by autophagy in the infected cells	Promotion on viral RNA replication Repression of innate antiviral immunity	Kim et al. [434]

**Table 2 ijms-19-03940-t002:** Summary of DENV-autophagy interactions.

Genotype /Serotype	Experimental Model	Characteristics of Autophagy	Functional Target	Reference
DENV-2 (PL046)	Human hepatocellular carcinoma, Huh7 cells/Infection The baby hamster kidney, BHK cells/Infection Mouse embryonic fibroblast, wild type and ATG5 knockout cells/Infection	Immunofluorescence detection of GFP-LC3-labeled punctate structure in the infected cells Elevated the conversion of LC3-I to LC3-II in the infected cells Induction of autophagosome fusion with lysosome in the infected cells Transmission electron microscopy observation of autophagic vacuoles in the infected cells Inhibition of virus production by 3-methyladenine (autophagy inhibitor) in the infected cells Increased virus production by rapamycin (autophagy inhibitor) in the infected cells Reduced virus production in the infected cells by knockdown of ATG5	Promotion on viral RNA replication	Lee et al. [398]
DENV-2 (16681)	Human monocytic cell line, U937 cells/Infection The rhesus monkey kidney, LLC-MK2 cells/Infection Human embryonic kidneys/SV40 large T antigen, HEK293T cells/Infection	Increased the conversion of LC3-I to LC3-II in the infected cells Induction of autophagosome fusion with lysosome in the infected cells Reduction of virus production in the infected cells by 3-methyladenine (autophagy inhibitor) No apparent effects on virus production by interference with autolysosome maturation in the infected cells Increased virus production in the infected cells by overexpression of a dominant mutant of PI-3K/Vps34 Activation of autophagy by ER stress in the infected cells	Restriction on virus production	Panyasrivanit et al. [462]
DENV-2 (16681)	Human hepatocellular carcinoma, Huh7, Huh7.5.1 and HepG2 cells/Infection The baby hamster kidney, BHK cells/Infection	Immunofluorescence detection of GFP-LC3-labeled punctate structure in the infected cells Induction of autophagosome fusion with lysosome in the infected cells Association between GFP-LC3-labled autophagic vacuoles and lipid droplets (LDs) in the infected cells Increased number of LDs in the infected cells by 3-methyladenine and knockdown of ATG12 and Beclin 1 Reduced virus replication by 3-methyladenine and knockdown of Beclin 1 in the infected cells Promotion on -oxidation of free fatty acids in the infected cells by autophagic degradation of LDs	Promotion on viral RNA replication Promotion on LDs catabolism	Heaton et al. [463]
DENV-2 (16681) DENV-3 (16562)	Human hepatocellular carcinoma, HepG2 cells/Infection The rhesus monkey kidney, LLC-MK2 cells/Infection	Increased the conversion of LC3-I to LC3-II in the infected cells Induction of autophagosome fusion with lysosome in the infected cells The colocalization of NS1 and dsRNA with autophagic vacuoles in the infected cells The colocalization between dsRNA and cathepsin D in the infected cells Reduction of virus production in the infected cells by 3-methyladenine (autophagy inhibitor) Inhibited virus replication in the infected cells by L-Asparagine (autolysosome inhibitor) Increased virus replication in the infected cells by rapamycin (autophagy inducer)	Promotion on viral RNA replication	Khakpoor et al. [464]
DENV-2 (16681)	Human hepatocellular carcinoma, HepG2 cells/Infection	Upregulation of LC3-I to LC3-II conversion in the infected cells Induction of autophagosome fusion with lysosome in the infected cells The colocalization of NS1, dsRNA, ribosomal L28 protein with LC3-labeled autophagic vacuoles in the infected cells Decreases in the intra- and extra-cellular virus amounts in the infected cells by 3-methyladenine (autophagy inhibitor) Slightly increased the intracellular and extracellular levels of viral particles by L-Asparagine (autolysosome inhibitor) Colocalization of mannose-6-phosphate and LC3 with dsRNA within amphisome in the infected cells	Support on viral RNA replication and translation	Panyasrivanit et al. [465,466]
DENV-2 (PL046)	ICR mice	Increase in the LC3-labeled autophagic vacuoles in the brain of infected mice Colocalization of NS1 with LC3-labeled punctate structure in the brain of infected mice Electron micrograph of autophagic vacuoles in the brain of infected mice Induction of amphisome formation Enhanced autophagic flux in the brain of infected mice Decreased the clinic score and increased survival rate of the infected mice by treating 3-methyladenine (autophagy inhibitor)	Promotion on virus replication and disease progression in vivo	Lee et al. [467]
DENV-2 (16681) DENV-2 (PL046)	Human hepatocellular carcinoma, Huh7.A.1 cells/Infection The baby hamster kidney, BHK cells/Infection AG129 mice (129/Sv mice lacking alpha/beta interferon [IFN-α/β] and IFN-γ receptors)/Infection	Inhibited the generation of mature virion in the infected cells by spautin-1 (autophagy inhibitor) Increase in the intracellular viral RNA in the infected cells by spautin-1 (autophagy inhibitor) Decreased survival rate of the infected AG129 mice	Promotion on virus replication Facilitation on the secretion of mature virion Promotion on disease progression in vivo	Mateo et al. [468]
DENV-2 (16681)	Human basophil precursor, KU812 cells/Infection Human immature mast, HMC-1 cells/Infection	Electron micrograph of autophagosome in the infected cells Enhanced autophagosome formation by antibody-dependent enhancement (ADE) Colocalization of E protein with LC3 by DENV ADE infection Colocalization of E protein and with autophagosome by DENV- and DENV ADE-infection Reduced DENV infection by overexpression of ATG4B^C74A^ dominant mutant	Facilitation of DENV ADE infection	Fang et al. [469]
DENV-1 (766733A) DENV-2 (PL046) DENV-3 (739079A) DENV-4 (4/H-241)	1. Human monocytic cell line, U937 cells /Infection 2. Mouse embryonic fibroblast, wild type and ATG5 knockout cells/Infection	Decreased virus infection by deficiency of ATG5 in the infected cells Colocalization of virus particles and autophagy machinery in the infected cells Reduced infection viral particles in the infected cells by lack of autophagy Colocalization of E protein and with autophagosome by DENV- and DENV ADE-infection	Promotion on cell to cell transmission	Wu et al. [470]
DENV-2 (16681)	Human hepatocellular carcinoma, Huh7.5.1 cells/Infection	Increased the GFP-LC3-labeled punctate structure in the infected cells Engulfment of the infecting virion by autophagosome in the infected cells Inhibited virus replication and secretion of virion by 3-methyladenine (autophagy inhibitor) Increased virus production by rapamycin (autophagy inducer)	Facilitation on virus entry	Chu et al. [471]
DENV-2 (NGC)	Human hepatocellular carcinoma, Huh7 cells/Infection	Block of the degradation of autophagic vacuoles and induction of autophagosome formation in the infected cells Reduction of autophagosome fusion with lysosome at the late stage of infection Promoted p62/SQSTM1 degradation through proteasomal pathway in the infected cells Suppression of virus replication in the infected cells by p62/SQSTM1	Promotion on virus replication	Metz et al. [472]
DENV-1 (Hawaii) DENV-2 (16681) DENV-2 (NGC) DENV-3 (H87) DENV-4 (Jamaique 8343)	Human hepatocellular carcinoma, HepG2 cells/Infection	Activation of lipophagy through Ancient ubiquitous protein 1 (AUP1) in the infected cells Induction of lipophagy by co-expression of NS4A and NS4B Inhibition of virus production by deficiency of AUP1-mediated lipophagy	Promotion on lipophagy Promotion on virus replication	Zhang et al. [473]
DENV-2 (16681)	Human hepatocellular carcinoma, HepG2 cells/Infection	Decreased virus production in the infected cells by silencing of the AMP-activated protein kinase alpha-1 (AMPKα1) Inhibition of viral-induced lipophagy by knockdown of (AMPKα1) Inhibited virus replication in the infected cells by selective inhibitor of (AMPKα1) Downregulation of virus replication by knockdown of Tuberous Sclerosis Complex 2 (TSC2) Activation of AMPK and inhibition of mTORC1 signaling in the infected cells	Promotion on lipophagy Promotion on virus replication	Jordan and Randall [474]
DENV-2 (M544)	The Madin-Darby Canine Kidney, MDCK cells/Infection	Protection the infected cells from death Induction of autophagy through PI-3K in the infected cells Increased LC3-I to LC3-II conversion in the infected cells Downregulation of the extracellular amount of virion by inhibition of autophagy Activation of autophagy by NS4A protein	Promotion on virus production	McLean et al. [475]
N/A	Human dermal microvascular endothelium, HMEC-1 cells/Infection	Increased the permeability of endothelial cells by NS1 protein Induction of autophagy by NS1 protein Impaired NS1-induced vascular leakage by inhibition of autophagy	Promotion on disease pathogenesis	Chen et al. [476]
DENV-2 (PL046)	Human hepatocellular carcinoma, Huh7 cells/Infection Human lung adenocarcinoma, A549 cells/Infection Mouse embryonic fibroblast, wild type and ATG5 knockout cells/Infection ICR mice	Induction of autophagy by activation of ER stress in the infected cells and mice Activation of unfolded protein response by virus infection Activation of autophagy by inositol-requiring enzyme 1α (Ire1α)/ c-Jun N-terminal kinases (JNK) signaling Reduced virus production by interfering with the Ire1α/JNK1-mediated autophagy Alleviation of disease symptoms and mortality rate in the infected mice by inhibiting JNK signaling	Promotion on virus production Promotion on disease pathogenesis	Lee et al. [477]
DENV-2	The Madin-Darby Canine Kidney, MDCK cells/Infection	Induction of ER stress in the infected cells Activation of unfolded protein response, protein kinase R (PKR)-like endoplasmic reticulum kinase (PERK) by virus infection Activation of autophagy by PERK-mediated signaling in the infected cells Requirement of Ataxia Telangiectasia Mutated (ATM) for activation of PERK and autophagy in the infected cells Inhibited virus production by interfering with UPR and autophagy	Promotion on virus production Promotion on survival rate of infected cells	Datan et al. [478]
DENV-2 (16681)	Human hepatocellular carcinoma, HepG2 cells/Infection	Induction ER stress in the infected cells Activation of unfolded protein response by virus infection Induction of cell apoptosis by virus infection	Possibly promotion on virus production Possibly promotion on survival rate of infected cells	Thepparit et al. [479]

**Table 3 ijms-19-03940-t003:** Summary of JEV-autophagy interactions.

Genotype /Serotype	Experimental Model	Characteristics of Autophagy	Functional Target	Reference
JEV (RP-9) strain JEV (RP-2m) strain	Human malignant pluripotent embryonal, NT-2 cells/Infection The baby hamster kidney, BHK cells/Infection	Increased the conversion of LC3-I to LC3-II in the infected cells No changes on autophagic flux in the infected cells Increased virus production by rapamycin (autophagy inducer) Inhibition of virus production by 3-methyladenine (autophagy Inhibitor) Reduced virus production by knockdown of ATG5 and Beclin 1 No significant colocalization between mcherry-LC3-labeled autophagic vacuoles and dsRNA replicating intermediate in the infected cells Apparent colocalization between the infecting particles with autophagic vacuoles in the infected cells	Facilitation on the virion entry and uncoating	Li et al. [480]
JEV (SA14-14-2) strain JEV (P3) strain	Mouse brain neuroblastoma, Neuro-2A cells/Infection The baby hamster kidney, BHK cells/Infection Human lung adenocarcinoma, A549 cells/Infection	Transmission electron microscopy observation of autophagic vacuoles in the infected cells Immunofluorescence detection of GFP-LC3-labeled punctate structure in the infected cells Elevated the conversion of LC3-I to LC3-II in the infected cells Induction of autophagosome fusion with lysosome in the infected cells Downregulation of virus production in the infected cells by interference with autolysosome maturation Abolished virus replication in the infected cells by knockout of ATG5 and Beclin 1 Increased cell death of the infected cells by knockout of ATG5 and Beclin 1 Upregulated interferon response in infected cells by gene silencing of ATG5 and Beclin 1	Promotion on viral RNA replication Protection from cell death Regulation of innate antiviral immunity	Jin et al. [481]
JEV (JaGAr-01) strain	Mouse embryonic fibroblast, wild type and p62/SQSTM1 knockout cells/Infection	Reduced the intracellular JEV viral RNA in the infected p62/SQSTM1 knockout cells Abolished the extracellular amount of JEV infectious particles in the infected p62/SQSTM1 knockout cells Impaired virus replication in the infected cells by lack of p62/SQSTM1	Promotion on viral RNA replication	Tasaki et al. [482]
JEV (SA14-14-2) strain	Mouse brain neuroblastoma, Neuro-2A cells/Infection The baby hamster kidney, BHK cells/Infection Human embryonic kidneys/SV40 large T antigen, HEK293T cells/Infection The pig kidney PK-15 cells/Infection	Transmission electron microscopy observation of autophagic vacuoles in the infected cells Increased the conversion of LC3-I to LC3-II in the infected cells Colocalization of NS1 with LC3 in the infected cells Enhanced virus replication by rapamycin (autophagy inducer) Reduced virus replication by 3-methyladenine (autophagy inhibitor) Decrease in virus replication by knockdown of ATG5 and ATG7 Induction of autophagy by C, NS1 and NS3 proteins through immunity-related GTPases M (IRGM)	Promotion on viral RNA replication	Wang et al. [483]
JEV (P20778) strain	Mouse brain neuroblastoma, Neuro-2A cells/Infection The porcine stable kidney, PS cells/Infection	Upregulation of LC3-I to LC3-II conversion in the infected cells Activation of autophagy through activating transcription factor 6 (ATF6)- and X-box binding protein 1 (XBP1)-mediated ER stress in the infected cells Inhibitions of ATG3 and Beclin expressions in the infected cells by knockdown of ATF6 and XBP1. Reduced virus replication in the infected cells by activating ER stress	Inhibition on virus infection	Sharma et al. [484]
JEV (SA14-14-2) strain	Human neuroblastoma, SK-N-SH cells/Infection Human neuroblastoma, SH-SY5Y cells/Infection Human umbilical vein endothelial, HUVECs/Infection Human hepatocellular carcinoma, Huh7/Infection The baby hamster kidney, BHK cells/Infection	Elevation of LC3-I to LC3-II conversion in the infected cells Increased virus production in the infected cells by knockdown of Beclin 1 Promotion of virus production in the infected cells by NEDD4-mediated suppression of autophagy	Inhibition on virus infection	Xu et al. [486]
JEV (P20778) strain	Mouse brain neuroblastoma, Neuro-2A cells/Infection The porcine stable kidney, PS cells/Infection The kidney epithelial cells extracted from an African green monkey, Vero cells/Infection Mouse embryonic fibroblast, wild type and ATG5 knockout cells/Infection	Elevation of LC3-I to LC3-II conversion in the infected cells Increased virus production in the infected cells knockout of ATG5 in the infected cells Upregulated virus production in the infected cells knockdown of ATG7 Enhanced cell death of the infected cells by interfering autophagy Colocalization of NS1 with LC3-labeled autophagic vacuoles and lysosomes in the infected cells Colocalization of NS1 and dsRNA within LC3-I-associated ER degradation enhancer, mannosidase a-like 1 (EDEM1) in the infected cells Abolishment of virus production in the infected cells by knockdown of LC3	Inhibition on virus production	Sharma et al. [487]

**Table 4 ijms-19-03940-t004:** Summary of WNV-autophagy interactions.

Genotype /Serotype	Experimental Model	Characteristics of Autophagy	Functional Target	Reference
382-99 (NY99) strain Kenyan strain of WNV	The kidney epithelial cells extracted from an African green monkey, Vero cells/Infection The hamster kidney, BHK cells/Infection Mouse embryonic fibroblast, wild type and ATG5 knockout cells/Infection	Transmission electron microscopy observation of autophagosome in the infected cells Immunofluorescence detection of GFP-LC3-labeled punctate structure in the infected cells Induction of autophagosomes fusion with lysosomes in the infected cells Inhibition of virus-induced autophagy by 3-methyladenine and gene knockout of ATG5 (autophagy inhibitor) No significant change of p62/SQSTM1 level by virus infection Reduced virus production by 3-methyladenine and wortmannin (inhibitors of initiating autophagy) No apparent effect on virus production in the infected cells depleted of ATG5	Promotion on viral RNA replication (specifically by the early stage of autophagy)	Beatman et al. [488]
WNV 6-LP strain WNV-MAD78 strain	The kidney epithelial cells extracted from an African green monkey, Vero cells/Infection Human embryonic kidneys/SV40 large T antigen, HEK293T cells/Infection Human hepatocellular carcinoma, Huh7 and Huh7.5.1 cells/Infection Human cervical cancer, HeLa cells/Infection Human lung adenocarcinoma, A549 cells/Infection Human brain cortical astrocyte/Infection Human foreskin fibroblast/Infection Mouse brain neuroblastoma, Neuro-2A cells/Infection	No effects on the LC3-I to LC3-II conversion in the infected 293T cells No significant change of p62/SQSTM1 in the infected 293T cells No interference with autophagy initiation in the infected 293T cells No apparent induction of autophagy in the infected Huh7, Huh7.5.1 and Neuro-2A cells No upregulation of autophagy in the infected primary human foreskin fibroblast and cortical astrocyte No significant inhibition on virus production by knockout of ATG5 and ATG7	Unknown	Vandergaast et al. [489]
WNV 6-LP strain	Human embryonic kidneys/SV40 large T antigen, HEK293T cells/Infection The kidney epithelial cells extracted from an African green monkey, Vero/Infection Human neuroblastoma, SK-N-SH cells/Infection Human cervical cancer, HeLa cells/Infection Mouse embryonic fibroblast, wild type and ATG5 knockout cells/Infection	Immunofluorescence detection of LC3-labeled autophagic vacuoles Increased conversion of LC3-I to LC3-II. Enhanced autophagic flux virus infection Decreased virus production by knockout of ATG5 Inhibited viral replication in the infected Hela cells that were treated by autophagy inducer, Tat-Beclin 1	Inhibition the viral genome replication at late infection stage	Kobayashi et al. [490]
WNV TX02 train	The 5-day-old C57BL/6J mice	Downregulated viral titer in the Tat-Beclin1 peptide-administrated mice Reduced the virus-induced mortality rate of mice by Tat-Beclin1 peptide	Inhibition of virus infection	Shoji-Kawata et al. [492]
WNV North American isolates	The kidney epithelial cells extracted from an African green monkey, Vero cells/Infection	The differential induction of autophagy between WNV variants Elevation of LC3-I to LC3-II conversion Modulated autophagy response by WNV NS4A and NS4B mutations Uncoupling of unfolded protein response to autophagy activation	Unknown	Blázquez et al. [493]

**Table 5 ijms-19-03940-t005:** Summary of ZIKV-autophagy interactions.

Genotype /Serotype	Experimental Model	Characteristics of Autophagy	Functional Target	Reference
PF-25013-18 (French Polynesia, 2013)	Primary human dermal fibroblasts/Infection	Transmission electron microscopy observation of autophagosome in the infected cells Immunofluorescence detection of LC3-labeled punctate structure in the infected cells Inhibition of viral replication in the infected cells by 3-methylaadenine (autophagy inhibitor) Increase in virus replication in the infected cells by Torin1 (autophagy enhancer)	Promotion on viral RNA replication	Hamel et al. [494]
MR766 (Uganda, 1947) H/PF/2013 (French Polynesia, 2013) IbH30656 (Nigeria, 1968)	Human fetal neural stem cells/Infection	Immunofluorescence detection of LC3-labeled punctate structure in the infected cells Elevation of LC3-I to LC3-II conversion in the infected cells Inhibition of viral replication of infected cells by 3-methylaadenine and chloroquine (autophagy inhibitor) Increase in virus replication of infected cells by Torin1 (rapamycin) Induction of autophagy by NS4A and NS4B Inhibition of Akt-mTOR signaling by virus infection and by NS4A and NS4B	Promotion on viral RNA replication	Liang et al. [495]
GZ01 (China, 2016)	Human umbilical vein endothelial cells/Infection	Immunofluorescence detection of mTagRFP-mWasabi-LC3-labeled autophagic vacuoles in the infected cells Increased conversion of LC3-I to LC3-II in the infected cells Degradation of p62/SQSTM1 in the infected cells Abolished virus production of infected cells by wortmannin and chloroquine (autophagy inhibitor) Enhanced virus production of infected cells by rapamycin Inhibition of virus production in the infected cells by knockdown of Beclin 1	Promotion on viral RNA replication	Peng et al. [496]
The Brazilian strain (Paraiba, 2015)	Human cytotrophoblast cell line, JEG-3/Infection *Atg16l1* HM mice/Infection	Accumulation of GFP-LC3-labeled punctate structure in the infected cells Increase in LC3-I to LC3-II conversion in the infected cells Downregulation of virus production of infected cells by 3-methyladenine, chloroquine and bafilomycin A1 (autophagy inhibitor) Enhancement of virus production in the infected cells by rapamycin and torin1 Impaired the transmission from maternal to fetal in mice by ATG16L knockout Decrease in the maternal-fetal transmission by hydroxychloroquine	Promotion on viral RNA replication Enhancement of utero transmission	Cao et al. [497]
MR766 (Uganda, 1947)	Drosophila melanogaster/Infection	Induction of LC3-I to LC3-II conversion in the brain of fly Increased the foci of mcherry-tagged ATG8 in the brain of fly Increased virus production by knockdown of ATG5 Activation of antiviral response by NF-kB and dSTING	Restriction of virus infection by activation of antiviral immunity	Liu et al. [498]

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
