# Peer review of "The Multifaceted Roles of Autophagy in Flavivirus-Host Interactions"

_ijms, 2018, doi:10.3390/ijms19123940_

Reviewer 1 Report
Po-Yuan Ke appropriately reviewed the roles of autophagy in flavivirus-host interaction and made reference to sufficient number of available literatures on this subject. This article gives in depth consideration to various available literatures and well describes the author′s knowledge on the role of autophagy in the Hepacivirus and flaviviruses life cycle and the impacts of virus induced autophagy on the pathogeneses. However, the article seems to discuss little about the potential use of autophagy as a therapeutic target for curing flavivirus infections and related human diseases.
Minor comments:
1. In line 164, consider changing the title “Flaviviruses” to include Hepacivirus within its meaning.
2. In line 204 to 205, please explain “the uncertain and controversial effects of DAA treatment”.
3. In line 379, consider changing “flaviviruses, among which HCV and DENV” to “flaviviridae, among which HCV and DENV”.
4. In line 407, “Huh7.5-1 cells “ should be revised to “Huh7.5.1 cells”.
Author Response
Dear reviewer:
Thank you for giving us the opportunity to resubmit our manuscript entitled “The Multifaceted Roles of Autophagy in Flavivirus-Host Interactions” to the International Journal of Molecular Sciences (Manuscript ID: ijms-397284). We are very grateful for the thoughtful and constructive suggestions raised by the reviewers. The content of this manuscript has been improved based on the reviewers’ comments, and section 5 has incorporated a discussion on the potential use of autophagy modulation as a curative therapy for flavivirus infection. The changes are shown in the revised manuscript, and point-by-point responses to each comment are listed as the follows.
To Reviewer 1:
Point 1: In line 164, consider changing the title “Flaviviruses” to include Hepacivirus within its meaning.
Response 1: We fully agree with the reviewer’s suggestion to change the title so that it more appropriately includes Hepacivirus. In the revised manuscript, changed the title from “Flaviviruses” to “Flaviviridae viruses” (page 8, line 164).
Point 2: In line 204 to 205, please explain “the uncertain and controversial effects of DAA treatment”.
Response 2: Thank you for these comments. We have extended our manuscript to briefly introduce the current controversy on the treatment outcome associated with the risk of HCC occurrence in HCV-patients after DAA-induced sustained virological response (SVR). Although DAA therapy has attained a cure rate of more than 90% in most HCV infections, an increase in HCC development was unexpectedly reported by several different groups. Following these studies, the data from other study cohorts showed no evidence for the promoting effect of DAA in HCC over the treatment course. However, the absolute risk of HCC still remains high in HCV patients with established liver cirrhosis. Please see lines 206-212 of paragraph 2 on page 9 in the revised manuscript.
Point 3: In line 379, consider changing “flaviviruses, among which HCV and DENV” to “flaviviridae, among which HCV and DENV”.
Response 3: We agree with the reviewer’s thoughtful suggestions. The term “flaviviruses” has been changed to “Flaviviridae”. Please see line 385 on page 12 in the revised manuscript.
Point 4: In line 407, “Huh7.5-1 cells” should be revised to “Huh7.5.1 cells”.
Response 4: Thank you for this comment. We have revised “Huh7.5-1” to “Huh7.5.1”. Please see line 413 of paragraph 2 on page 13 in the revised manuscript.
We hope that this version of our manuscript and our responses to the reviewers meet the criteria for publication in the International Journal of Molecular Sciences. Thank you for your kind consideration.
Reviewer 2 Report
The author presents a very detailed summary of the role of autophagy during flavivirus infections. This review will be valuable to members of both the autophagy and flavivirus fields. The only thing lacking, is a section on the potential use of autophagy as a therapeutic target during flavivirus infection. To further enhance the review article, I have provided a few comments below to help with ease of reading.
Section 2.1: This section should be broken up for ease of reading. On line 26, I think ‘on the other hand”, should be “At the same time”. Based on the information in the text, this complex seems to form at the same time as the trimeric complex to help form the phagophore.
Line 33. A new paragraph should star with the talk of fusion of autophagosomes.
Section 3: There is a lot of information on the different viruses that probably isn’t necessary unless it is relatable to autophagy. This could be combined with section 4 or each virus could get its own section if they are different from each other.
Tables: This is an impressive summary of all of the work done with HCV and the other flaviviruses and autophagy. However, it is cumbersome and some of the same material is listed in each line. I wonder if rearranging by functional target would be better because the author also mentions the outcomes of each study in the text. A more concise summary may be useful for readers.
In the abstract, the author mentions the potential use of autophagy as a therapeutic target. I found this section to be hard to find in the text. If the author wants to include this line. A separate section before the conclusions should be added to discuss the potential benefits of targeting autophagy.
Line 52: font issues here and throughout the text.
Lines 79-80, 177 and others: Underlined text??
Line 222: heparin should be heparan
Line 261: DNEV should be DENV
Author Response
Dear reviewer:
Thank you for giving us the opportunity to resubmit our manuscript entitled “The Multifaceted Roles of Autophagy in Flavivirus-Host Interactions” to the International Journal of Molecular Sciences (Manuscript ID: ijms-397284). We are very grateful for the thoughtful and constructive suggestions raised by the reviewers. The content of this manuscript has been improved based on the reviewers’ comments, and section 5 has incorporated a discussion on the potential use of autophagy modulation as a curative therapy for flavivirus infection. The changes are shown in the revised manuscript, and point-by-point responses to each comment are listed as the follows.
To Reviewer 2:
Point 1: This section should be broken up for ease of reading. On line 26, I think ‘on the other hand”, should be “At the same time”. Based on the information in the text, this complex seems to form at the same time as the trimeric complex to help form the phagophore.
Response 1: We appreciate the reviewer’s suggestions and have revised this section into two paragraphs: lines 3-33 on page 5 in a paragraph and lines 34-54 on pages 5-6 as a new paragraph. In addition, the term “on the other hand” was changed to “At the same time”. Please see line 26 of paragraph 1 on page 5 in the revised manuscript.
Point 2: A new paragraph should star with the talk of fusion of autophagosomes.
Response 2: Thank you for this comment. A new paragraph has been included starting from the content regarding the fusion of autophagosomes with lysosomes. Please see line 34 of paragraph 2 on page 5 in the revised manuscript.
Point 3: Section 3: There is a lot of information on the different viruses that probably isn’t necessary unless it is relatable to autophagy. This could be combined with section 4 or each virus could get its own section if they are different from each other.
Response 3: We are very grateful for the reviewer’s thoughtful suggestions on section 3. In the past decade, several studies have demonstrated the functional roles of autophagy in flavivirus infection. However, large discrepancies and controversial conclusions were observed among different studies, which may have been related to the use of different types of approaches and viral infection contexts. Some studies analyzed the function of autophagy in the context of a complete infecting cycle, whereas other groups utilized the replicon viral RNA and/or viral proteins that contain a piece of a fragment in the viral genome. Also, different genotypes of flaviviruses used in different studies also led to divergent functions of autophagy in flavivirus infections. In addition, autophagy was shown to affect different infection stages of the virus infection and lead to different outcomes during viral-induced pathogenesis. Therefore, we intend to provide compelling information on the history of the discovery and identification of genotypes, the complete viral life cycle, the interactions with host cells, and the pathogenesis of viral infection in section 3. The information in section 3 will provide readers with a brief background of each flavivirus, and it describes the unanswered questions associated with viral infection as well as unresolved issues for treatment. The information in section 3 will help readers to understand and compare the different models used for autophagy in section 4. Introducing the virology of flaviviruses before reading section 4 allows for a more comprehensive vision on the interactions between flaviviruses and autophagy, and possible reasons for the disparities among different studies in section 4 are proposed. We hope that the reviewer agrees with the separation of sections 3 and 4 in the revised manuscript. Thank you again for your constructive comments.
Point 4: This is an impressive summary of all of the work done with HCV and the other flaviviruses and autophagy. However, it is cumbersome and some of the same material is listed in each line. I wonder if rearranging by functional target would be better because the author also mentions the outcomes of each study in the text. A more concise summary may be useful for readers.
Response 4: We appreciate the reviewer’s comments on the tables. Most previous review articles on virus autophagy focus on briefly introducing the conclusions of different studies and limited information on the detailed methods and few comparisons between different papers were provided. We expect that this review article will provide comprehensive information on the conclusions from different approaches and cell contexts in summative tables. The information in these tables will also allow the readers to easily compare the infecting genotypes used, the methods utilized for characterizing autophagy, the different functional impacts of the conclusions regarding autophagy on viruses and hosts, and the potential effects of autophagy modulation on viral growth and host responses. I have reformatted these tables in the revised manuscript to better represent the data and clarify the information. Thank you again for the thoughtful suggestions.
Point 5: In the abstract, the author mentions the potential use of autophagy as a therapeutic target. I found this section to be hard to find in the text. If the author wants to include this line. A separate section before the conclusions should be added to discuss the potential benefits of targeting autophagy.
Response 5: Thank you for these thoughtful suggestions on the discussion of the potential benefits of targeting autophagy in the treatment of flavivirus infection. We have extended our manuscript to cover the therapeutic potential of modulating autophagy for curing flavivirus infection in section 5 in the revised manuscript. The potential obstacles in the development of autophagy modulators, particularly for inhibitors for clinic medicine to cure flavivirus infection, are also discussed in this section. Please see lines 1-31 of paragraph 1 on page 36 in the revised manuscript.
Point 6: Line 52: font issues here and throughout the text.
Response 6: Thank you for this comment. We have edited this mistake caused by errors in the quotation marks, which are incompatible in different computer systems. Please see line 52 of paragraph 2 on page 2 in the revised manuscript. Other mistakes associated with quotation marks have been corrected throughout this manuscript.
Point 7: 79-80, 177 and others: Underlined text??
Response 7: We apologize for these mistakes in formatting. These mistakes have been corrected. The unusual underlined characters in the text have been removed. Please see lines 78-79 of paragraph 3 on page 6, line 176 of paragraph 3 on page 8, lines 277-278 of paragraph 4 on page 10, line 29 of paragraph 2 on page 20, and line 22 of paragraph 2 on page 33 in the revised manuscript.
Point 8: Line 222: heparin should be heparan.
Response 8: Thank you for this reminder. We have edited “heparin” to “heparan”. Please see line 228 of paragraph 4 on page 9 in the revised manuscript.
Point 9: Line 261: DNEV should be DENV
Response 9: We apologize for the spelling errors. We have checked the spelling and edited “DNEV” to “DENV” in the text. Please see line 267 and line 282 of paragraph on page 10, line 146 of paragraph 4 on page 22 in the revised manuscript.
We hope that this version of our manuscript and our responses to the reviewers meet the criteria for publication in the International Journal of Molecular Sciences. Thank you for your kind consideration.